

**1** **The wet deposition of the inorganic ions in the 320 cities across**

**2** **China: spatiotemporal variation, source apportionment, and**

**3** **dominant factors**

**4** Rui Li[a], Lulu Cui[a], Yilong Zhao[a], Ziyu Zhang[a], Tianming Sun[a], Junlin Li[a], Wenhui

**5** Zhou[a], Ya Meng[a], Hongbo Fu[a,b,c] *

**6** *[a] Shanghai Key Laboratory of Atmospheric Particle Pollution and Prevention, Department of*

**7** *Environmental Science & Engineering, Institute of Atmospheric Sciences, Fudan University,*

**8** *Shanghai, 200433, P.R. China*

**9** *[b] Shanghai Institute of Pollution Control and Ecological Security, Shanghai 200092, P.R. China*

**10** *[c] Collaborative Innovation Center of Atmospheric Environment and Equipment Technology*

**11** *(CICAEET), Nanjing University of Information Science and Technology, Nanjing 210044, P.R.*

**12** *China*

**13** **Corresponding author**

**14** fuhb@fudan.edu.cn

**15** **Abstract**

**16** The acid deposition has been considered to be a severe environmental issue in China. The pH,

**17** electrical conductivity (EC), and the concentrations of the water soluble ions ($NO_3^-$, $Cl^-$, $Ca^{2+}$, $K^+$,

**18** $F^-$, $NH_4^+$, $Mg^{2+}$, $SO_4^{2-}$, and $Na^+$) in the precipitation samples collected from the 320 cities during

**19** 2011-2016 across the whole China were measured. The mean concentrations of $F^-$, $NO_3^-$ and $SO_4^{2-}$

**20** were in the order of winter (6.10, 19.44 and 45.74 µeq/L) > spring (3.45, 13.83, and 42.61 µeq/L) >

**21** autumn (2.67, 9.73, and 28.85 µeq/L) > summer (2.04, 7.66, and 19.26 µeq/L). The secondary ions

**22** ($SO_4^{2-}$, $NO_3^-$ and $NH_4^+$), and $F^-$ peaked in Yangtze River Delta (YRD) and Sichuan basin (SB). The



crustal ions (i.e., $Ca^{2+}$, $Mg^{2+}$), $Na^+$, and $Cl^-$ showed the highest concentrations in the semi-arid
regions and the coastal cities, respectively. The statistical methods confirmed that the mean
anthropogenic contribution ratios to $SO_4^{2-}$, $F^-$, $NO_3^-$, and $NH_4^+$ at a national scale were 46.12%,
71.02%, 79.10%, and 82.40%, respectively. However, $Mg^{2+}$ (70.51%), $K^+$ (77.44%), and $Ca^{2+}$
(82.17%) were mostly originated from the crustal source. Both $Na^+$ (70.54%) and $Cl^-$ (60.42%) were
closely linked to the sea-salt aerosols. On the basis of the stepwise regression (SR) analysis, it was
proposed that most of the secondary ions and $F^-$ were closely related to gross industrial production
(GIP), total energy consumption (TEC), vehicle ownership, and N fertilizer use, but the crustal ions
($Ca^{2+}$ and $K^+$) were mainly controlled by the dust events. The influence of dust days, air temperature,
and wind speed on ions increased from Southeast China (SEC) to Central China, and then to
Northwest China (NWC), whereas the influence of socioeconomic factors on acid ions ($SO_4^{2-}$ and
$NO_3^-$) displayed the higher value in East China.
**Keywords:** Water-soluble ions; precipitation; spatiotemporal variation; source identification; China
**1.   Introduction**

Atmospheric wet deposition generally removes efficiently the aerosol particles and dissolved

gaseous pollutants from the atmosphere (Garland, 1978; Al-Khashman, 2005; Migliavacca et al.,
2005). However, in some regions with severe air pollution, the scavenging of substantial aerosol
particles alters the chemical compositions of precipitation and even aggravates the acid deposition
(Kuang et al., 2016). Some inorganic ions (i.e., $SO_4^{2-}$, $NO_3^-$, $NH_4^+$, $Ca^{2+}$) play significant roles on
the terrestrial and aquatic ecosystem via wet deposition; for instance, leading to severe soil (lake)
acidification (alkalization), inhibiting the plant growth, and changing the regional climate (Liu et
al., 2011; Yan et al., 2010; Larssen and Carmichael, 2000; Larssen et al., 1999). In the past decades,



China has been suffered from the severe air pollution along with the population growth and
industrialization (Liu et al., 2016a). Therefore, the investigation of the wet deposition status of
inorganic ions is of great interest to the public and policy makers (Négrel et al., 2007).

A large amount of studies mainly focused on the spatiotemporal variation of the S and N

deposition around the world due to their adversely ecological effects in the past decades (Gerson et
al., 2016; Clemens 2006; Zhang et al., 2010). Okuda et al. (2005) showed that the $SO_4^{2-}$
concentration in the precipitation exhibited a slight decrease coupling with the decrease of the $SO_2$
concentration in Tokyo during 1990-2012. Hunová et al. (2014) reported that the averagely S
deposition flux decreased from 181 kg/ha/year to 100 kg/ha/year in Czech during 1995 and 2011 on
the basis of the data in 15 cities. Du et al. (2012) estimated that the wet deposition flux of inorganic
nitrogen reached 3.5 kg N/ha/year according to the average of 151 monitoring in the United States
during 1985-2012, which were significantly lower than that of China during the same period (11.11-
13.87 kg/ha/yr) (Jia et al., 2014).

Many researches about the S and N deposition have been extensively performed to date in China

in the recent years (Jia et al., 2014; Xu et al., 2015). In the past decades, the anthropogenic emissions
of $SO_2$, $NO_2$, and $NH_3$ displayed the remarkable increase along with the dramatic increase of fossil
fuel and fertilizer consumption in China (Jia et al., 2014; Kuribayashi et al., 2012). It was well
documented that the gaseous precursors containing S and N could be transformed into sulfates
($SO_4^{2-}$), nitrates ($NO_3^-$), and ammonium ($NH_4^+$) during ageing in the atmosphere, thereby
contributing to the formation of airborne fine particles, of which were considered to be the main
reason for the persistent fog and haze pollution in China (Wang et al., 2016a; Qiao et al., 2015). At
a city level, Huang et al. (2008) observed that the wet deposition fluxes of $SO_4^{2-}$, $NH_4^+$, and $Ca^{2+}$



displayed the slight decrease from 1986 to 2006 in the urban of Shenzhen, whereas the wet
deposition of $NO_3^-$ increased rapidly during the same period. Very recently, Pu et al. (2017) reported
that the $SO_4^{2-}$ concentration in the wet deposition of Shangdianzi (a regional background station of
Beijing) showed slight decrease during 2003-2014, but the $NO_3^-$ concentration showed an opposite
trend. At a regional scale, Pan et al. (2013) observed that the highest S wet deposition was
concentrated in the urban and industrial region of Tianjin among of ten sites of North China (NC).
Song et al. (2017) suggested that the bulk deposition fluxes were in the order of Chengdu (urban) >
Yanting (agricultural area) > Gongga mountain (natural reserve). At a national scale, Jia et al. (2014)
firstly found that the wet deposition of N in Southeast China (SEC) showed a significant decrease,
whereas it increased slightly in the western of China on the foundation of the data (620 monitoring
sites) collected from 120 cities across China during 1990 and 2010. Following this work, Liu et al.
(2016) further observed that the serious S deposition (79 monitoring sites) on SEC and Southwest
China (SWC). In these studies, the spatial distributions of both S and N were determined using the
spatial interpolation method, which generally required substantial monitoring sites (city > 150, and
monitoring site > 300). However, these conclusions were obtained based on a small quantity of
monitoring sites, which increased the uncertainties of the results. Meanwhile, the monitoring sites
in these studies were mainly located on some remote regions such as mountain or rural site rather
than the mixture of urban, suburban, and rural sites, which cannot accurately reflect the spatial
variations of inorganic ions in China. Moreover, the spatiotemporal variations of other inorganic
ions (i.e., $K^+$, $Ca^+$, $Mg^{2+}$) remained unclear to date, which were also linked to the acid deposition,
as well as the haze pollution in China (Mikhailova et al., 2013; Aloisi et al., 2017; Müller et al.,

2015).





Based on these field measurements, the ion levels in the deposition across China were believed
to be underestimated due to the few ion species measured by previous studies (Liu et al., 2016a),
which was closely associated with various emission sources (Kuang et al., 2016). Thus, the source
identification should be performed to assess accurately their contributions to the wet deposition
(Larssen et al., 1999). Liu et al. (2015b) identified that the $Cl^-$ and $NH_4^+$ in the precipitation of Tibet
were both originated from the marine and crustal source using the geochemical index method. On
the basis of the positive matrix factorization (PMF) model, Qiao et al. (2015) showed that fossil fuel
combustion and agriculture were the main sources of $SO_4^{2-}$ and $NO_3^-$ in Jiuzhaigou (Sichuan
province). In a newly work reported by Leng et al. (2018), they supposed that the combustion of
fossil fuels, domestic sewages, and fertilizers were the main sources of the N-bearing ions on the
basis of the N isotope analysis. To date, some methods, including geochemical index method,
multivariate analyses, and isotope signatures have been utilized to identify the anthropogenic versus
natural sources of the inorganic ions in the precipitation. However, these methods suffered from
some weaknesses from different standpoints (AlKhatib and Eisenhauer 2017; Shi et al., 2014). For
instance, the geochemical index methods cannot estimate the contribution ratios of multiple sources
to $Ca^{2+}$ and $Na^+$ at a spatial scale (Liu et al., 2015b). Despite the advances of multivariate analyses
lowering the associated uncertainties, the multi-collinearity still disturbed the predictions of these
models (Shi et al., 2014). The isotope signature method was costly and complex, especially for the
unconventional stable isotopes (i.e., K, Ca) (AlKhatib and Eisenhauer 2017), which restricted its
application at a large scale. Therefore, multiple source apportionment methods should be combined
in order to enhance the reliability of the results. Liu et al. (2015) also demonstrated that the
geochemical index method coupled with multiple statistics decreased the uncertainties of results.





Apart from the source apportionment, the key factor identification for the ions in the wet
deposition is also of great importance to reduce the acid deposition. At an early study, Singh and
Agrawal (2008) revealed that the significant increase of vehicle emissions contributed to the
accumulation of $NO_2$, which might be an important precursor of acid rain. Allen et al. (2015)
observed that some inland cities in arid and semi-arid regions were generally subjected to dust
events, which could increase the $Ca^{2+}$ and $K^+$ concentrations in the wet deposition. Following this
work, Yu et al. (2017a) found that considerable energy consumption, gross domestic production
(GDP), and emitted substantial pollutants made China as major regions of acid rain around the world
using path analysis and correlation analysis. However, these researches only assessed the limited
factors for the inorganic ions in the wet deposition (Yu et al., 2016; Yu et al., 2017a), ignoring the
contributions of other socioeconomic and natural factors. Moreover, these researches mainly
focused the whole effects of the influential factors on inorganic ions at a national scale, while they
did not consider the spatial heterogeneity of the influential factors, resulting possibly in the great
deviation of the inorganic ions in the wet deposition for the different regions.
Here, the data of nine water-soluble ions in the precipitation including $Ca^{2+}$, $Cl^-$, $F^-$, $K^+$, $Mg^{2+}$,
$Na^+$, $NH_4^+$, $NO_3^-$, and $SO_4^{2-}$ in the 320 cities across the whole China were collected during 2011-
2016 to examine the characteristics of the main water-soluble ions in the precipitation. Specifically,
the objectives of our study were (1) to reveal the spatiotemporal patterns of water-soluble ions in
the precipitation recently in China at a national scale; (2) to identify quantitatively the source of the
water-soluble ions in the precipitation based on the multiple statistical methods; and (3) to seek out
the key factors for the inorganic ions at a spatial scale. This study supplied the systematical data for
comprehensive understanding on the inorganic composition in the precipitation based on the long-




term field measurement, at a national scale (the 1282 monitoring sites distributed in the 320 cities
across the whole China), which was beneficial to the implementation of appropriate strategies to
promote environmental protection in China.
**2.   Materials and methods**
2.1  Site description

The spatial distribution of field stations in National Acid Deposition Monitoring Network

(NADMN) is illustrated in Fig. 1. The selected 1282 monitoring sites are distributed in the 320 cities
across 31 provinces. These cities are classified into Northeast China (NEC), NC, SEC, Northwest
China (NWC), and Southwest China (SWC) (Tab. S1). Both of NEC and NC show typical
temperature monsoon climate, while SEC presents the subtropical monsoon climate. The SWC
region suffers from the combined effects of subtropical monsoon climate and tropical monsoon
climate. NWC suffers from the temperate continental climate and displays minor rainfall amount.
NEC and NC are filled with temperature deciduous forest, whereas SEC is mainly occupied by the
subtropical evergreen forest. The subtropical evergreen forest and tropical evergreen forest spread
out the SWC region. The NWC is generally filled with expansive grasslands and desert. The
latitudes and longitudes of all of 1282 monitoring sites range from 18.25 to 50.78° $N$, and from 79.57
to 129.25° $E$, respectively. Annual mean rainfall ranges from 10 to 1853 mm and the annual mean
air temperature varies between -6.9 and 24.3 ℃. The monitoring sites were designed as a mixture
of urban and background sites. Most of these sites are concentrated in urban region, and a few of
sites in suburban and rural areas are considered as the background sites.
2.2  Sampling and chemical analysis

The real-time precipitation was collected by monitors in the field stations as a routine procedure



of NADMN. Samples from each monitoring site were collected using wet deposition automatic
collectors (diameter 30 cm) installed at 1.5 m above ground level. The cover of the collection
instrument opened automatically without delay when the precipitation sensor was activated and
closed automatically when precipitation ceased and no water remained on the sensor surface. After
the sampling, the pH and EC values of the samples were measured immediately. The sample pH
was measured using a pH meter (MP-6p, HACH, USA) at 20–25°C. The EC value of the
precipitation samples was determined by an EC meter (CyberScan, CON1500, USA). After the
analysis of pH and EC, all of the samples were contained in the pre-cleaned polyethylene plastic
bottles at -18°C in order to prevent the possible transformation by microbes. All of the plastic
buckets and the polyethylene plastic bottles were cleaned with deionized water for more than three
times and then air-dried in clean room prior to use.

All of the precipitation samples were used to analyze the concentrations of the water-soluble

ions including $NO_3^-$, $Cl^-$, $Ca^{2+}$, $K^+$, $F^-$, $NH_4^+$, $Mg^{2+}$, $SO_4^{2-}$, and $Na^+$. The microporous membranes
(0.45 μm) were employed to remove all of insoluble particulates (< 0.45μm) from the precipitation
samples before the analysis. The ion concentrations were determined through ion chromatography
(Dionex ICS-900) equipped with a conductivity detector (ASRS-ULTRA). The CS12A column and
AS11-HC column were applied to determine the cations and anions, respectively. Each sample was
measured for more than three times and the relative standard deviation was less than 5% for each
ion. Analysis of the blank samples once a month confirmed that the cross contamination in the
present research was negligible. For each ion, the analysis of simulated precipitation suggested that
the relative bias was lower than 10%.
2.3  Data calculation





The monthly and annual volume-weighted mean (VWM) concentrations were calculated based
on the concentrations of specific ions and precipitation. The monthly and annual VWM
concentrations were obtained as follows:
$$C_x = \frac{\sum_{i=1}^{n}(C_i(x) \times P_i)}{\sum_{i=1}^{n} P_i} \qquad (1)$$

where $C_x$ denoted the monthly and annual VWM concentration of the given ion; $C_i(x)$ was the
concentration of the given ion in the precipitation (μeq/L); $P_i$ was the precipitation in individual
sample. The monthly and annual VWM pH values were obtained based on the corresponding VWM
concentrations of H$^+$ via Eq. (1).
The wet deposition flux of the given ion was calculated using the following Eq. (2)
$$D_w = P_t C_w / 100 \qquad (2)$$

where $D_w$ was the wet deposition flux of the given ion (kg N ha$^{-1}$); $P_t$ was the total amount of the
precipitation events (mm); $C_w$ was the VWM concentration of each ion (mg/L); and 100 was a unit
conversion factor.
In order to obtain the contributions of various alkaline species to acid neutralization in the
precipitation, the neutralization factor (NF) was calculated using the following Eq. (3)-(5)
(Kulshrestha et al., 1995):
$$NF_{NH_4^+} = \frac{NH_4^+}{NO_3^- + SO_4^{2-}} \qquad (3)$$

$$NF_{Ca^{2+}} = \frac{Ca^{2+}}{NO_3^- + SO_4^{2-}} \qquad (4)$$

$$NF_{Mg^{2+}} = \frac{Mg^{2+}}{NO_3^- + SO_4^{2-}} \qquad (5)$$





2.4 Source apportionment of ionic species in wet deposition
The enrichment factor (EF) has been widely applied to estimate the contribution ratios of the
various sources to the major ions in the previous studies (Lawson and Winchester 1979; Cao et al.,
2009; Lu et al., 2011). In the present study, an ion EF in the precipitation relative to the ion in the
sea was calculated using Na as a reference element as follows:
$$EF_{sea} = \frac{(X / Na^+)_{precipitation}}{(X / Na^+)_{sea}} \quad (6)$$

where $EF_{sea}$ was the enrichment indicator of a given ion in the precipitation relative to the ion in the
sea; $X$ was the ion in the precipitation; $(X/Na^+)_{precipitation}$ represented the ratio of components in the
precipitation; $(X/Na^+)_{sea}$ denoted the ratio of components in the sea (Keene et al., 1986; Turekian,

1968).

The EF value of an ion in the precipitation relative to the corresponding ion in the soil was
calculated following Eq. (7):
$$EF_{soil} = \frac{(X / Ca^{2+})_{precipitation}}{(X / Ca^{2+})_{soil}} \quad (7)$$

where $EF_{soil}$ represented the EF value of an ion in the precipitation relative to the corresponding ion
in the soil; X denoted an ion in the precipitation; $(X/Ca^{2+})_{precipitation}$ was the ratio of components in
the precipitation; $(X/Ca^{2+})_{sea}$ denoted the ratio of components in the soil (Wei et al., 1991; Wei et al.,
1992; Shi et al., 1996; Zhang et al., 2012; Chen et al., 1992).
In order to quantify the anthropogenic source versus natural one of ionic species in the
precipitation. The fractions of anthropogenic, marine, and crustal source contributed to the ions in
the precipitation were calculated as follows:



$$SSF = \frac{(X/Na^+)_{sea}}{(X/Na^+)_{precipitation}} \times 100\% \quad (8)$$

$$CF = \frac{(X/Ca^{2+})_{soil}}{(X/Ca^{2+})_{rain}} \times 100\% \quad (9)$$

$$AF = 100\% - SSF - CF \quad (10)$$

where $SSF$ represented the fraction of sea salt; $CF$ denoted the crustal contribution; and $AF$ denoted
the anthropogenic fraction. $SSF$ was recalculated as the difference between 1 and $CF$ when $SSF$ was
greater than 1; $CF$ was recalculated as the difference between 1 and $SSF$ when $CF$ was higher than

1.

FA has been widely employed to determine the contribution ratios of natural and anthropogenic

source to ionic species in the precipitation. First of all, FA was applied to reduce the dimension of
original variables (measured ion concentrations in samples) and to extract a small number of
principal components to analyze the relationships among the observed variables. All of the factors
with eigenvalues over 1 were extracted based on the Kaiser-Meyer-Olkin (KMO) test and the
Bartlett's test of sphericity, and were rotated using the Varimax method. The FA factor scores and
each ion concentration were treated as independent and dependent variables, respectively. The
resultant regression coefficients were employed to convert the absolute factor scores and then to
calculate the contribution of each PC source (Luo et al., 2015).
2.5 The geographical weight regression (GWR) method

Although the relationships between the independent variables and the dependent variables could

be calculated using correlation analysis and multiple linear regression analysis (MLR), these
methods cannot show the spatial variability of regression coefficients. Thus, the GWR method was
applied to generate the local regression coefficients for each city, which were then mapped to display





the spatial variability. Local regression coefficients were obtained using weighted least squares with
the following weighting function (Brunsdon et al., 1996):
$$\beta(u_i, v_i) = (X^{\mathrm{T}} W(u_i, v_i) X)^{-1} X^T W(u_i, v_i) Y \quad (11)$$
where $\beta(u_i, v_i)$ represented the local regression coefficient at city i; $X$ was the matrix of the
influential factors; $Y$ denoted the matrix of the wet deposition fluxes of the water-soluble ions; and
$W(u_i, v_i)$ was an n order matrix that the diagonal elements were the spatial weighting of the influential
factors. The spatial weight function was calculated via the exponential distance decay form:
$$W(u_i, v_i) = \exp(-d^2(u_i, v_i)/b^2) \quad (12)$$
where $d(u_i, v_i)$ represented the distance between the location i and j, and b was the kernel bandwidth.
2.6 Data source and statistical analysis

The data of GDP, gross industrial production (GIP), N fertilizer use, vehicle ownership, urban

green space (UGS) during 2011-2016 were collected from China City Statistical Book. Total energy
consumption (TEC) during the period were obtained from China Energy Statistical Yearbook, which
consisted of the consumption of coal, crude oil, and natural gas. The daily meteorological factors
including precipitation, maximum and minimum air temperature, wind speed, air pressure, relative
humidity (RH) during 2011-2016 were collected from China Meteorological Data Network. The
daily visibility data during 2011-2016 was collected from National Centers for Environmental
Prediction (NCEP). The data of dust days were calculated based on the horizon visibility data. The
days with the visibility lower than 1 km were treated as the dust days. The daily data of $PM_{2.5}$, $PM_{10}$,
$SO_2$, and $NO_2$ were downloaded from the National Environmental Monitoring Platform
(https://www.aqistudy.cn/historydata/). These data at a national scale were open access since
January 2014. To match the meteorological data at a national scale, the data of air pollutants during



2014-2016 were applied to investigate the relationships of the water-soluble ions, meteorological
factors, and the air pollutants in the atmosphere (Tab. S2). In addition, the SR analysis was employed
to determine the key factors regulating the wet deposition fluxes of the water-soluble ions. All of
the statistical analysis were performed by the software package of ArcGIS 10.2, SPSS 21.0, and
Origin 8.0 for Windows 10.
**3 Results and discussion**
3.1 The pH and EC values in the precipitation

To obtain the preliminary knowledge about the precipitation characteristics, the basic

physiochemical properties including pH and EC of the precipitation samples are presented in Fig.
2. The annually pH during 2011 and 2016 ranged from $5.45 \pm 0.27$ (mean ± standard deviation) to
$5.94 \pm 0.46$ and the mean value was 5.76 (Fig. 2a). Seinfeld (1986) estimated that the precipitation
with pH lower than 5.60 was considered as acid rain because the pH value of natural water in
equilibrium with atmospheric $CO_2$ was 5.60. However, the $CO_2$ level has been increasing in recent
years and thus the equilibrium pH has changed (McGlade and Ekins 2015) Therefore, the average
$CO_2$ concentration during 2011-2016 (396.83 ppm) around the world was applied to the present
study (http://www.ipcc.ch/). The ionization equation of $CO_2$ include $CO_2+H_2O=H_2CO_3$ and
$H_2CO_3=HCO_3^-+H^+$. The dissociation constant of two equations are $3.47\times10^{-2}$ ($K_0$) and $4.4\times10^{-7}$ ($K_1$),
respectively. The $(c(H+))^2 = K_0\times K_1\times P_{CO2} = 6.06 \times 10^{-12}$. Therefore, the equilibrium pH was 5.61,
which was slightly higher than the current value (pH = 5.60). Herein, 41% of the samples during
the measurement showed the pH value below 5.61. Compared with the pH value of the precipitation
during 1980-2000 (Wang and Xu 2009), the pH value of the precipitation showed a remarkable
increase in recent years. For instance, the pH value in the precipitation of SWC increased from 3.5-





4.0 (the mean value of 1980-2000) to 5.87 during 2011-2016. Although some cities in Hunan and
Hubei province (e.g., Chengzhou, Erzhou) still suffered from the severe acid deposition, the mean
pH values (4.46) of the two provinces during 2011-2016 were slightly higher than those in 1980-
2000 (3.5-4.0). It was well known that precipitation pH was associated with the $SO_2$ and $NO_x$
emissions (Pu et al., 2017). Due to the implementation of $SO_2$ control measurements since the 11th
Five-year Plan, the $SO_2$ column concentration over China displayed a marked decrease after 2007
based on Global Ozone Monitoring Experiment (GOME), reported by Gottwald and Bovensmann
(2011). Based on the bottom-up method, Liu et al. (2010) also supposed that $SO_2$ emission began to
decrease since 2007 (Lu et al., 2010), in good agreement with the results obtained from the remote
sensing. Besides, nearly all of the power plants built newly and the in-use plants have been required
to be equipped with advanced selective catalytic reduction (SCR) or selective non-catalytic
reduction (SNCR) since 2010 (Tian et al., 2013; Lu et al., 2011), resulting in a gradual decrease of
the   $NO_x$   emission   after   2010   (China   Statistical   Yearbook,
http://data.stats.gov.cn/easyquery.htm?cn=C01). Based on the result of correlation analysis (Tab.
S2), the pH value showed the significantly negative correlation with $SO_2$ and $NO_2$ in the ambient
air especially with the increased RH.   Thus, it could be proposed that the pH value of the
precipitation in most of the regions of China during 2011 and 2016 were significantly higher than
those before 2000 due to the decreases of the $SO_2$ and $NO_x$ emissions.
The pH value in the precipitation at a national scale exhibited significantly seasonal variation
with the highest value in summer (6.57), followed by autumn (5.64), spring (5.49), and the lowest
one in winter (5.32) (Fig. 2b). The seasonal variation of pH values in wet deposition was supposed
to be linked with the wash-out effect of precipitation on atmospheric particular matters (Xing et al.,



303 2017), which was supported by the positive relevance between pH and precipitation ($p < 0.01$).

304 Besides, the scavenging atmospheric $SO_2$ by precipitation may also play an important role in the

305 seasonal variation of the pH values (Wu and Han, 2015). The atmospheric $SO_2$ concentration was

306 the lowest in summer and the highest in winter. The highest atmospheric $SO_2$ and sulfate

307 concentrations in winter of the north part of China were partially ascribed to the intensive domestic

308 coal combustion for heating (Liu et al., 2016b; Liu et al., 2017).

309  At a spatial scale across the whole China (Fig. 3a), the pH value of the precipitation presented a

310 gradual increase from SEC to NC and NWC. The relatively low pH values in the precipitation were

311 usually observed in YRD (i.e., Huzhou, Ningbo, and Shanghai), Hunan province (i.e., Changde,

312 Changsha, and Loudi), Hubei province (i.e., Wuhan), and Jiangxi province (i.e., Nanchang, Yichun,

313 and Jingdezhen), but the relatively high pH values occurred in NC and NWC, especially in Xinjiang

314 autonomous region (i.e., Changji, Altai, Urumqi and Aksu). Among of the 320 cities, the lowest one

315 and the highest one were located in Huzhou, (3.20, Zhejiang province), and Altai, (6.82, Xinjiang

316 autonomous region), respectively (Fig. 3). Compared with high acidity in some cities of SEC, the

317 acidity of the precipitation in many cities of NC could be largely neutralized by some alkaline ions

318 because the saline-alkali soils were widely distributed in NC (Wang et al., 2014). Some city

319 atmosphere (i.e., Urumqi and Altay) in Xinjiang autonomous region were frequently attacked by

320 local continental dust particles, diluting the precipitation acidity (Rao et al., 2015).

321  The annually mean EC varied from $10.18 \pm 3.21$ μS cm$^{-1}$ to $13.33 \pm 3.75$ μS cm$^{-1}$ during the

322 period (Fig. 2a). The EC value was mainly affected by total water-soluble ions in the precipitation

323 and rainfall amount, of which indirectly reflected the cleanliness of the precipitation and the air

324 pollution status. The decrease of EC in recent years suggested that air pollution in China has been





mitigated due to the implementation of special air pollution control measures (Wang et al., 2017;
Yang et al., 2016). The EC value also presented distinctly seasonal variation and showed the highest
value in spring (Fig. 2c), followed by ones in summer and autumn, and the lowest one in winter,
which was apparently different from the seasonal pH variation. Among all of the inorganic ions,
only $Ca^{2+}$ displayed notable relationship with EC ($p < 0.01$). It was supposed that many crustal ions
such as $Ca^{2+}$ could be lifted up and transported to East China by frequent dust storms in spring and
summer, thereby leading to the high EC value in the precipitation (Fu et al., 2014). The mean EC
value exhibited a significantly spatial variation with the higher ones in Shizuishan (36.60 μS cm⁻¹)
and Yinchuan (24.79 μS cm⁻¹) (Ningxia autonomous region), Wuwei (60.01 μS cm⁻¹) (Gansu
province), Edors (28.72 μS cm⁻¹) (Inner Mongolia autonomous region), and Aksu (22.06 μS cm⁻¹)
(Xinjiang autonomous region) and the lower one in some remote regions such as Lhasa (3.42μS cm⁻
¹) (Tibet autonomous region), Aba (2.20 μS cm⁻¹) (Sichuan province) and Diqing (2.46) (Yunan
province) (Fig. 3b). The lowest and highest EC were observed in Aba (2.20 μS cm⁻¹) and Wuwei
(60.01 μS cm⁻¹), respectively (Fig. 3). The cities in the western and northern of Sichuan province,
and the southern of Tibet autonomous region presented the lower EC values due to the sparse
population and minimal industrial activity. Although TB has received the effects of the industrial
emissions and biomass burning from South Asia via a long-range atmospheric transport, most of the
pollutants tended to be deposited on the South of Himalayas except persistent organic pollutants
(POPs) (Yang et al., 2016b; Dong et al., 2017). The cities with higher EC was generally close to the
Taklamakan and Gobi deserts. Strong winds in these deserts stirred a large amount of dusts, and
then caused many dust events, resulting in high loading of $Ca^{2+}$ and $Mg^{2+}$ (Wang et al., 2016d). The
positive relationship between wind speed and EC also revealed that strong wind promoted the





accumulation of crustal ions over China (Tab. S2).
3.2  Chemical composition in the precipitation
3.2.1    The inter-annual variation of the water-soluble ions

The inter-annual variation of the ionic constitutes of the precipitation in China during 2011-2016

are summarized in Fig. 4. The concentrations of $Na^+$, $NO_3^-$, and $SO_4^{2-}$ increased from 7.26 ± 2.51,
11.56 ± 3.71, and 33.73 ± 7.59 μeq/L to 11.04 ± 4.64, 13.59 ± 2.63, and 41.95 ± 8.64 μeq/L during
2011 and 2014, respectively (Fig. 4a). However, $Na^+$, $NO_3^-$, and $SO_4^{2-}$ concentrations decreased
from the highest ones in 2014 to 9.75 ± 2.89, 12.29 ± 4.02, and 30.57 ± 7.43 μeq/L in 2016. The
concentrations of $Ca^{2+}$, $NH_4^+$, and $Mg^{2+}$ increased from 31.59 ± 8.29, 14.84 ± 4.63, and 8.77 ± 2.42,
to 58.84 ± 10.31, 41.33 ± 10.26, and 10.49 ± 3.07 during 2011-2013 (Fig. 4a), whereas they
decreased from the peak values in 2013 to 31.20 ± 8.48, 18.13 ± 4.84, and 8.93 ± 2.92 μeq/L in
2016, respectively. The $F^-$ concentration exhibited gradual decrease from 3.63 to 2.96 μeq/L during
2012-2016. However, the $K^+$ and $Cl^-$ concentration fluctuated during 2011 and 2016 and did not
display regularly annual variation.

It was well documented that the $SO_4^{2-}$ concentration was closely associated with the $SO_2$

emissions because $SO_2$ in the ambient air could be transformed into $SO_4^{2-}$ during aging in the
atmosphere (Qiao et al., 2015). In the present study, $SO_4^{2-}$ in the precipitation exhibited a marked
correlation with $SO_2$ in the ambient air ($p < 0.01$), especially with the increased RH (Tab. S2). The
total $SO_2$ emissions in China decreased dramatically due to the installation of the flue gas
desulfurization (FGD) systems and the closure of less efficient power plants in China since 2012
(Li et al., 2017b). At a national scale, the remarkable decrease of the $SO_4^{2-}$ concentration was
observed since 2014, which lagged behind the decrease of the $SO_2$ emission. Such scenario was



widely observed in some developed countries such as Japan (Okuda et al., 2005). However, some
cities (i.e., Beijing and Baoding) in NC showed the notable decreases since 2012, which
corresponded to the decrease of the total $SO_2$ emission. It was supposed that the electrostatic
precipitators (ESP) and fabric filters (FFs) for the sulfates removal were more widely applied to
steel and iron plants, and cement production process, both of which were widely distributed in NC
(Hua et al., 2016; Wang et al., 2016b). Moreover, coal has been gradually replaced by natural gas
for domestic heating in Beijing, resulting in the less $SO_2$ emission and thus decreasing the $SO_2$
concentration in the ambient air (Pu et al., 2017). Based on the open data downloaded from National
Environmental Monitoring Platform, the annually mean $SO_2$ concentration in Beijing decreased
from 22.0 μg/m$^3$ to 9.29 μg/m$^3$ during 2014-2016, in good agreement with the temporal variation of
$SO_4^{2-}$ in the precipitation.

The $NO_x$ emission decreased rapidly after the upgrading of oil product quality standards, the

import denitrification facilities, and the implementation of low-$NO_2$ burner technologies (Li et al.,
2016; Liu et al., 2017). However, the $NO_3^-$ concentration in the precipitation over China only
displayed slight decrease during this period. It was assumed that the high $NO_3^-$ in the precipitation
resulted from the increase of motor vehicles (Link et al., 2017). Based on the bottom-up method,
the estimated $NO_x$ emissions from vehicle exhausts in China linearly increased by 75% since 1998
(Wu et al., 2016). Shandong suffered from the highest vehicle emissions among all of the provinces,
of which the $NO_x$ released from vehicle exhausts in Shandong province increased from 477.6 Gg to
513.8 Gg during 2011-2014 (Sun et al., 2016), corresponding to the annual variation of $NO_3^-$ in the
precipitation of Jinan and Linyi. The $NO_3^-/SO_4^{2-}$ value was recognized as an important index to
determine the relative importance of nitrate (mobile) vs. sulfate (stationary) emission in the





atmosphere (Arimoto et al., 1996). The value of $NO_3^-/SO_4^{2-}$ at the national scale was still lower than
1, suggesting that the contribution of sulfate to the acidity of the precipitation was still higher than
that of $NO_3^-$. Nevertheless, the ratio in the precipitation showed a gradual increase from 0.33 to 0.40
during this period, indicating that the precipitation type in China has evolved from sulfuric acid type
to a mixed type controlled by sulfuric and nitric acid.
The $NH_4^+$ level in the precipitation was closely linked to the $NH_3$ emission because $NH_3$ tended
to be neutralized to form $(NH_4)_2SO_4$ and $NH_4NO_3$ in the atmosphere (Zhang et al., 2016). The
anthropogenic emission of $NH_3$ was mainly derived from fertilizer use, livestock manures, vehicle
exhausts, and industrial processes (Kang et al., 2016). Wherein, livestock manures and synthetic
fertilizer application were considered as two major source of the $NH_3$ emission, accounting for 80-
90% of total emission (Kang et al., 2016; Xu et al., 2016). The nitrogen fertilizer consumption has
decreased since 2013 (http://www.stats.gov.cn/), which was in good agreement with the variation of
the $NH_4^+$ concentration in the precipitation. Therefore, the fertilizer consumption could be treated
as an important factor for the $NH_4^+$ level in the precipitation. However, the $NH_3$ emission from
livestock manures estimated by Kang et al. (2016) showed an opposite variation to the $NH_4^+$ level
in the precipitation collected herein. It was probably attributed to the slight decrease of air
temperature in the major cities of China during 2011-2013 because the actual $NH_3$ emission to the
atmosphere was sensitive to air temperature (Kang et al., 2016), which has been proved by the
correlation analysis (Tab. S2). Apart from the contribution source mentioned above, soil served as
major natural sources of the $NH_3$ emissions (Sun et al., 2014). Teng et al. (2017) demonstrated that
urban green space made a great contribution to the $NH_3$ amount in the atmosphere. In the present
study, the urban green space in some cities such as Lianyungang (Jiangsu province) and Qingdao



(Shandong province) showed the marked correlation with the $NH_4^+$ level in the wet deposition.

The long-range transport of dust aerosol was considered as the major source of $Ca^{2+}$ and $Mg^{2+}$

in the atmosphere (Fu et al., 2014). Song et al. (2016) reported that the magnitude of dust emissions
in spring generally decreased in the past decades. The dust deposition and ambient $PM_{10}$
concentration in the Xinjiang autonomous region also decreased dramatically during 2000-2013
(Zhang et al., 2017a). Here, $Ca^{2+}$ and $Mg^{2+}$ in the wet deposition of some cities such as Aksu in
Xinjiang autonomous region decreased from 32.37 to 4.80 μeq/L and from 15.80 to 4.81 μeq/L
during 2011-2016, respectively, corresponding to the decrease of dust deposition. However, the
decrease of $Ca^{2+}$ and $Mg^{2+}$ over China significantly lagged behind the reduction of dust deposition.
It was well known that the increase of soil particles and dusts due to urbanization might induce the
high level of $Ca^{2+}$ and $Mg^{2+}$ in the wet deposition (Lyu et al., 2016). The road mileage in China
increased by 25% from 2011 to 2013, while it only showed slight increase (2.52%) during 2013-
2016 (http://www.stats.gov.cn/). Padoan et al. (2017) also demonstrated that the resuspension of
road dust generally showed the highest impact on the emission of the Ca and Mg elements among
non-exhaust sources (i.e. tire wear, brake wear, road dust).

Both of $K^+$ and $Cl^-$ were identified as the important tracers for biomass burning and fireworks

(Cheng et al., 2014). Nevertheless, the $K^+$ and $Cl^-$ concentration in the precipitation did not reflect
the contribution of biomass burning because biomass burning usually occurred in dry seasons (Zhou
et al., 2017b). Furthermore, the $K^+$ concentration in the precipitation showed significantly
relationship with crustal ions ($Ca^{2+}$ (r = 0.40, p < 0.01) and $Mg^{2+}$ (r = 0.49, p < 0.01)) (Tab. S2),
suggesting that other sources could play important role on the accumulation of $K^+$ and $Cl^-$. Chen et
al. (2017b) recommended that fugitive dust to be the main source of $K^+$ when the mitigation



measures were seriously implemented. The minor F$^-$ in the wet deposition served as an indicator of
coal combustion because fluorine was generally released from coal combustion (Chen et al., 2013).
Recently, the F$^-$ emission displayed remarkable decrease because more coal-fired power plants were
equipped with FGD and dust removal equipment (Zhao and Luo, 2017), which explained the
decrease of F$^-$ in the precipitation of some industrial cities such as Baoding (3.22 to 1.65 during
2012-2016), Shijiazhuang (3.18 to 2.73), and Handan (3.88 to 3.53) in Hebei province. Na$^+$ was
generally originated from the transport of sea salt aerosols, fugitive dusts, and the incineration of
wastes and fossil fuels (Zhao et al., 2011). The Cl$^-$/Na$^+$ value in the precipitation of some coastal
cities (i.e. Lishui (1.15), Jiaxing (1.20), Dandong (1.18), Wenzhou (1.18)) were similar to the marine
equivalent Cl$^-$/Na$^+$ ratio (1.17) (Wang et al., 2015a), suggesting that Na$^+$ in the precipitation of these
coastal cities might be derived from ocean. However, the Cl$^-$/Na$^+$ ratios in the precipitation of some
regions far from the ocean were significantly higher than marine equivalent Cl$^-$/Na$^+$ ratio due to the
contribution of coal combustion (Liu et al., 2016b; Liu et al., 2017).
3.2.2 The seasonal variation of the inorganic ions in the wet deposition
The mean concentrations of SO$_4^{2-}$, NO$_3^-$ and F$^-$ in the wet deposition were in the order of winter
(SO$_4^{2-}$, NO$_3^-$ and F$^-$: 45.74, 19.44 and 6.10 μeq/L) > spring (42.61, 13.83, and 3.45 μeq/L) > autumn
(28.85, 9.73, and 2.67 μeq/L) > summer (19.26, 7.66, and 2.04 μeq/L) (Fig. 4b). It was well known
that SO$_4^{2-}$ and NO$_3^-$ were usually generated via the oxidation of SO$_2$ and NO$_2$ in the atmosphere,
respectively (Yang et al., 2016). The combustion of fossil fuels for domestic heating in winter
probably promoted the accumulation of SO$_2$ and NO$_2$ in the atmosphere (Liu et al., 2017; Lu et al.,
2010). Some cities in the NC region including Shijiazhuang and Zhengzhou showed the higher SO$_4^{2-}$
and NO$_3^-$ levels in the precipitation of winter compared with those in summer, which were in




agreement with the seasonal variation of $SO_2$ and $NO_2$ concentrations in the ambient air. It reflected
that the combustion of fossil fuels for domestic heating contributed to the accumulation of $SO_4^{2-}$ and
$NO_3^-$ and these ions deposited via the rainfall. Moreover, stagnant meteorological conditions
including shallow mixing layers, high atmospheric pressure, low precipitation, and low wind speed
occurred frequently in winter, thereby trapping more pollutants and elevating the concentrations of
$SO_2$ and $NO_2$ in the atmosphere (Tai et al., 2010). In contrast, strong solar radiation and turbulent
eddies from ocean in summer could promote the dispersion of these pollutants (Antony Chen et al.,
2001). For instance, some coastal cities such as Beihai (Guangxi autonomous region) and Haikou
(Hainan province) were generally exposed of strong solar radiation and high wind speed, which
significantly decreased the $SO_4^{2-}$ and $NO_3^-$ concentrations in the precipitation of summer (Beihai:
$SO_4^{2-}$ (6.06) and $NO_3^-$ (7.37); Haikou: $SO_4^{2-}$ (5.33) and $NO_3^-$ (4.96)). The $F^-$ concentration in the
precipitation displayed the similarly seasonal variation to $SO_4^{2-}$ and $NO_3^-$, which was likely
associated with the higher coal consumption for domestic heating in some industrial cities of NC,
NWC, and NEC (Ding et al., 2017).

The concentrations of $Cl^-$, $Ca^{2+}$, $K^+$, $NH_4^+$, $Mg^{2+}$, and $Na^+$ exhibited the highest values in summer,

followed by those in spring and autumn, and the lowest one in winter. The higher concentration of
$NH_4^+$ in the precipitation collected in summer was probably linked to agricultural activities. The
widespread utilization of fertilizer in summer have been observed over China (Zhang et al., 2011;
Tao et al., 2016), which could increase the $NH_3$ emission. In addition, the $NH_3$ emission was
sensitive to the air temperature and generally increased with the temperature (Kang et al., 2016).
The $NH_3$ released from agricultural activities could transform to $NH_4^+$, especially under the
condition of high RH (Li et al., 2013). Thus, the high $NH_3$ emission and rapid photochemical



reaction contribute to the higher $NH_4^+$ in the precipitation in summer. However, $K^+$, $Ca^{2+}$, and $Mg^{2+}$
displayed higher concentrations in spring and summer, which was probably related to the high
loading of fugitive dusts (Zhang et al., 2017c). Lyu et al. (2016) demonstrated that the high
temperature coupled with strong wind caused the lower water content in the road, leading to higher
tendency of dust re-suspension in the Wuhan summer. In the present study, these crustal ions in the
precipitation also showed the higher values in the summer of Wuhan. The high concentration of $Na^+$
and $Cl^-$ in spring and summer was probably attributed to the evaporation of sea salt under the
condition of high air temperature (Grythe et al., 2014). It was found that $Na^+$ in summer were 5.1-
10.3 times of those in winter in some coastal cities such as Qingdao (5.96) (Shandong province),
Qinhuangdao (9.65) (Hebei province), and Sanya (6.83) (Hainan province).
3.2.3 Spatial distribution of the water-soluble ions across the whole China

At a spatial scale, the annual mean concentrations of $NO_3^-$, $Cl^-$, $Ca^{2+}$, $K^+$, $F^-$, $NH_4^+$, $Mg^{2+}$, $SO_4^{2-}$,

and $Na^+$ ranged from 0.20 to 47.98 µeq/L, from 0.27 to 80.86 µeq/L, from 0.59 to 157.15 µeq/L,
from 0.15 to 23.43 µeq/L, from 0.11 to 11.64 µeq/L, from 0.20 to 84.24 µeq/L, from 0.28 to 39.30
µeq/L, from 0.29 to 191.95 µeq/L, and from 0.15 to 39.50 µeq/L during 2011-2016, respectively.
All of these water-soluble ions displayed significantly spatial variation, as shown in Fig. 5 and Fig.

6.

The mean concentrations of the secondary ions ($NO_3^-$, $NH_4^+$, and $SO_4^{2-}$) showed the highest

values in YRD (Changzhou (34.53, 73.40, and 80.47 µeq/L) (Fig. 5a-c) and Nanjing (35.62, 17.12,
and 49.51 µeq/L) and SB (Chengdu (38.08, 65.19, and 57.16 µeq/L) and Leshan (25.32, 38.99, and
61.24 µeq/L)), followed by ones in NC (Jinan (11.67, 16.57, and 58.28 µeq/L) and Anyang (20.46,
41.32, and 22.01 µeq/L), and the lowest ones in TB (0.50, 0.91, and 1.44 µeq/L) (Lhasa). Many



secondary ions exhibited the high concentrations in YRD because of intensive energy consumption
and industrial activities (Zhou et al., 2017a). For instance, the total energy consumption of the
Jiangsu province was second to Hebei province among all of the provinces in China (Wang 2014).
The $SO_2$ and $NO_x$ emissions from cement plants and iron and steel industries in Jiangsu and Zhejiang
province were significantly higher than those in other provinces (Hua et al., 2016; Wang et al.,
2016b), which was in coincident to the spatial agglomeration of the $SO_2$ and $NO_2$ concentrations in
the ambient air of these provinces It has been reported that the acid deposition pattern have moved
from SWC to SEC since 2000s (Yu et al., 2017a). However, SB still possessed high concentrations
of secondary ions in the precipitation because of high S content in the local consumed coals (Ren et
al., 2006). Besides, the unique topographic conditions and unfavorable diffusion conditions
facilitated the deposition of regionally transported pollutants stuck by Qinling mountains and Daba
mountain (Kuang et al., 2016), although the energy consumption of Sichuan province was much
less than those in other provinces (Tian et al., 2013). Moreover, the steady increase use of fertilizer
and livestock manures coupled with high air temperature made SB to be one of the $NH_3$ emission
hotspots (Li et al., 2017a). Nevertheless, some remote areas in NWC and SWC such as Lhasa and
Aba showed the lower secondary ions due to sparse population and anthropogenic activities (Li et
al., 2007). In these regions, these secondary ions were mainly derived from crustal source, and then
deposited concurrently in the rainfall events (Niu et al., 2014). Besides, relatively extensive
anthropogenic activities such as increased vehicle exhaust might promote the emissions of
secondary ions in the tourist season (Qiao et al., 2017). For instance, the number of tourists in Lhasa
have been increasing to 11 million until 2015 (http://www.xinhuanet.com/fortune/2016-
01/13/c_1117763885.htm), which could boost the slight increase of secondary ions in the wet





deposition.

F⁻ showed the higher concentrations in NC, YRD, and SB because many coal-fired power plants

and iron and steel industries were mainly concentrated in the Hebei and Jiangsu province (Liu et al.,
2015a) (Fig. 6a). Besides, Hebei and Jiangsu were two provinces with much higher coal
consumptions (Li et al., 2017), which could release large quantity of F⁻ to the atmosphere. Although
the power plants and iron and steel industries were relatively scarce in SB, many large phosphorite
mines might increase the F⁻ concentration in the precipitation (Wu et al., 2014). As one of the largest
phosphorite mine over China, Jinhe phosphorite mine was close to Chengdu, which significantly
increased the F⁻ concentration in the precipitation of Chengdu (9.21 μeq/L). Moreover, the high
abundance of F⁻ in the local coal (Mianyang: 269.25 μg/g, Guangan: 1061μg/g) also contributed to
the F⁻ emissions (Dai and Ren, 2006; Wang et al., 2016c; Ren et al., 2006). In addition, the F⁻ in the
precipitation showed remarkable relevance with $T_{max}$ based on the correlation analysis (r = 0.12, $p$
< 0.05). The annually mean air temperature in SB (17.2 °C) were slightly higher than that in Hebei
(14.3 °C) and Jiangsu (16.4 °C) province, thereby boosting the F⁻ emission.

The high concentrations of Cl⁻ were mainly concentrated on coastal cities such as Shanghai,

Lianyungang (Jiangsu province) and Qingdao (Shandong province) (Fig. 6b), indicating the effect
of sea-salt sourced from the ocean (Gu et al., 2011; Allen et al., 2015; Grythe et al., 2014). The high
Na⁺ concentration not only focused on these coastal cities (Fig. 6c), but also enrich in some arid and
semi-arid cities such as Jinchang (35.08 μeq/L) and Gannan (25.51 μeq/L) (Gansu province). It was
assumed that the windblown dust originated from Taklimakan Desert could play a vital role on the
enrichment of Na⁺ in Inner Mongolia and Hexi corridor because these regions were located on the
downwind direction of dust (Engelbrecht et al., 2016). Meanwhile, the evaporation of salt lakes in



West China might promote the $Na^+$ enrichment in the precipitation (Bian et al., 2017). Besides, the
dust event also promoted the elevation of $Ca^{2+}$, especially in Jiayuguan and Guyuan (Gansu province)
(Fig. 6d), both of which were located in the Hexi corridor (Allen et al., 2015). The $Mg^{2+}$ presented
higher value in some cities (Handan: 36.63 μeq/L, Liupanshui: 39.30 μeq/L) in the Hebei province
and Guizhou province (Fig. 6e). The soil in the Guizhou province possessed the highest Mg
concentration (843.33 mg/kg) in China (Li et al., 1992), where the $Mg^{2+}$ stored into the soils could
be lifted into the atmosphere by strong wind coupled with severe stony desertification (Jiang et al.,
2014). Although the Mg concentration in the soil of Hebei province was slightly lower compared
with those of Guizhou province, the bioavailable Mg concentration peaked in Hebei province (Hao
et al., 2016), which could be inclined to re-suspend into the atmosphere and then deposit with the
rainfall in the warm season.
3.2.4 Neutralization capacity of the alkaline ions

In order to reveal the most important ion for neutralization ($Ca^{2+}$, $NH_4^+$, and $Mg^{2+}$) in the

precipitation, the relative proportion of three NFs in all of the cities are summarized in Fig. 7. The
triangular diagram showed that the contribution of three ions were in the order of $Ca^{2+}$ (51.84%) >
$NH_4^+$ (34.14%) > $Mg^{2+}$ (14.02%). The NF ratios of $NH_4^+$ and $Ca^{2+}$ in China displayed the highest
values in summer, followed by ones in spring and autumn, and the lowest one in winter (Fig. 7a). It
was supposed that strong acid neutralization were mainly brought about by the alkaline ions via
high rainfall. Besides, the neutralization capacity of the alkaline ions reached higher in spring due
to the effects of dust events (Wang et al., 2015b). In the present study, the NFs of $NH_4^+$ and $Ca^{2+}$ in
Beijing ($NH_4^+$: 0.57, $Ca^{2+}$: 0.17) and Baoding ($NH_4^+$: 0.56, $Ca^{2+}$: 0.19) showed the markedly higher
values in spring. Zhai and Li (2003) also observed that most frequent dust storms generally occurred





in NC in spring. However, the NFs of $Mg^{2+}$ (0.70) showed the highest one in winter. Aside from the
temporal difference of neutralization, the NFs presented a significantly spatial variation in China
(Fig. 7b). The high NFs of $Ca^{2+}$ were mainly concentrated on some cities in NWC such as
Bayingolin (0.57) because these arid and semi-arid regions were exposed of periodic Asian dust
intrusions (Yu et al., 2017b). In the case of the typical dust events, the content of crustal species
such as Ca increased substantially (Chen et al., 2015). Compared with the other regions, the NFs of
$NH_4^+$ showed the higher value in some cities of SWC such as Chengdu (0.55). Kang et al. (2016)
demonstrated that the $NH_3$ emissions in Sichuan province were significantly higher than those in
other provinces of China, accounting for more than 10 % of the total emission from livestock
manures. The NFs of $Mg^{2+}$ peaked in NC, which was in good agreement with the higher
concentration of $Mg^{2+}$ in the wet deposition of NC. The higher concentration of bioavailable $Mg^{2+}$
in the soil was beneficial to increase the neutralization capacity of $Mg^{2+}$ in the wet deposition (Hao
et al., 2016), although the $SO_2$ and $NO_2$ emissions in NC were significantly higher than those in
other regions (Fu et al., 2016).
3.3  Comparisons of pH, EC, and the inorganic ion concentrations with the previous studies

The annual mean pH, EC and the inorganic ion levels in the precipitation of some metropolitans

across China are summarized in Tab. 1. The mean pH values of the most cities in SEC and SWC
(i.e., Shanghai: 4.39 and Wuhan: 4.68) were lower than those in some remote areas such as
Jiuzhaigou (5.95) and Yulong mountain (5.94) (Qiao et al., 2018; Niu et al., 2014), while the average
pH values of some cities in NC and NWC such as Zhengzhou (6.09) and Urumqi (6.13) were slightly
higher than those in remote areas. It was assumed that the remote areas were less affected
anthropogenic source except local tourist activities, while high aerosol emissions were mainly



centered on some metropolitans of SEC and SWC. The pH of the precipitation in Zhengzhou (pH =
6.09) (Henan province) and Urumqi (pH = 6.13) (Xinjiang autonomous region) showed high value
compared with some remote regions because of the strong neutralization capacity of alkaline ions
(Wang et al., 2014). Besides, the pH values in the wet deposition of most metropolitans in China
were also lower than those in some developing countries (e.g., Guaiba: 5.92, Petra: 6.80) (Tab. 1).
It was supposed that $SO_2$ and $NO_x$ emitted from industrial and vehicle emissions in China could be
higher than those in some countries such as Brazil and Jordan (Wu and Han 2015). In addition,
higher abundance of the neutralizing components in Jordan tended to increase pH of the
precipitation. On the other hand, the pH values of the wet deposition in most cities of China were
significantly higher than those in some cities of developed countries such as Sardinia (pH = 5.18)
(Italy) and Adirondack (pH = 4.50) (United States). It was assumed that many Western countries
were faced up with severe acid issue due to the rapid industrialization before 2002 (Sickles II and
Shadwick 2015). In addition, the annually mean rainfall amount in some cities of East China were
higher than those in Sardinia and Adirondack, which could dilute the acidity of the precipitation
(Tsai et al., 2011). The mean EC in the wet deposition of most cities over China were approximate
to those in some remote regions (i.e., Yulong Mountain, Jiuzhaigou), and some foreign cities such
as Guaiba, Brazil. However, Lanzhou (EC = 58.06 μS cm$^{-1}$) (Gansu province) and Petra (EC = 160
μS cm$^{-1}$) (Jordan) showed remarkably higher value than other cities, suggesting that the dust
cyclones from Taklamakan and Khamaseen played vital roles on the EC and chemical composition
in the precipitation (Abed et al., 2009).

The concentrations of $NO_3^-$, $SO_4^{2-}$, and $NH_4^+$ in the most cities of China except Qingdao

(Shandong province) and Lhasa (Tibet autonomous region) were significantly higher than those in



some natural reserve areas such as Jiuzhaigou, Yulong Mountain, and Nam Co (Qiao et al., 2018;
Niu et al., 2014) (Tab. 1), suggesting the local point and non-point emissions in these cities played
important roles on the concentrations of inorganic ions in the precipitation. However, the
concentrations of these inorganic ions in the most cities were lower than those in foreign cities such
as Singapore, Petra (Jordan), Tokyo, and Newark (United States) (Balasubramanian et al., 2001; Al-
Khashman et al., 2005; Okuda et al., 2005; Song and Gao 2009), indicating the effects of restricting
emissions of air pollutants since Chinese 12th Five-Year Plan (Liu et al., 2016a). However, some
cities including Shenyang (Liaoning province) and Chengdu (Sichuan province) were still faced up
with severe acid deposition. On the whole, the concentrations of the crustal ions ($Ca^{2+}$ and $Mg^{2+}$)
were in the order of the arid and semi-arid cities/regions (Nam Co, Urumqi, Lanzhou, and Petra) >
the inland cities and natural reserve regions (Chengdu and Yulong mountain) > the coastal cities
(i.e., Guaiba, Singapore, and Tokyo). Kang et al. (2016) reported that Tibetan Plateau have been
frequently affected by dust events under the condition of climate change in the past decades, which
probably increased the $Ca^{2+}$ and $Mg^{2+}$ levels in Nam Co. However, it should be noted that some
coastal cities such as Patras (Greece) and Sardinia (Italy) possessed higher $Ca^{2+}$ and $Mg^{2+}$ levels,
which was probably attributed to the long transport of the dust from of the Sahara desert (Kabatas
et al. 2014). Cabello et al. (2016) demonstrated that African air masses mostly reached some coastal
cities of Mediterranean on the basis of back-trajectory analysis.
3.4  The source apportionment of the ions in the precipitation across China
3.4.1 EF and geochemical index method

The mean values of EFs (seawater and soil), SSF and CF in all of the cities are listed in Tab. 2.

The water-soluble ion was treated to be enriched relative to the reference source when the EF value





of the ion was significantly higher than 1.00, whereas it was considered to be diluted when the EF
value of the ion was not much higher than 1.00. In the present study, the mean $EF_{sea}$ for $Na^+$, $Cl^-$,
$SO_4^{2-}$, $NH_4^+$, $K^+$, $Mg^{2+}$, $Ca^{2+}$, $NO_3^-$, and $F^-$ over China were 1.00, 1.13, 7.22, 10.51, 16.16, 18.18,
231.56, 3507.49, and 5864.28, suggesting that $Cl^-$ and $Na^+$ in the precipitation were enriched in the
marine origin at a national scale. The mean $EF_{soil}$ of $Mg^{2+}$, $K^+$, $Ca^{2+}$, $Na^+$, $SO_4^{2-}$, $F^-$, $NO_3^-$, $NH_4^+$, and
$Cl^-$ reached 0.55, 0.83, 1.00, 1.83, 5.13, 9.96, 59.36, 86.31, and 169.88, indicating that $Ca^{2+}$, $K^+$, and
$Mg^{2+}$ were considered to be originated from the crustal source. Both of the $EF_{sea}$ for $SO_4^{2-}$ and $NO_3^-$
showed significantly spatial variability and they presented the higher ones in YRD and SB
(significantly higher than 1) (Fig. 8a-b), which suggested that both of the ions were not mainly
sourced from the sea source. However, $EF_{sea}$ for $SO_4^{2-}$ in some cities such as Nujiang (0.92) and
Nanchong (0.81) were lower than 1. It was assumed that the Indian monsoon played an important
role on the wet deposition of $SO_4^{2-}$ (Gu et al., 2016). Except $SO_4^{2-}$ and $NO_3^-$, $EF_{sea}$ for other ions
showed relatively uniform distribution at a national scale. $EF_{sea}$ for $NH_4^+$, $F^-$, $Ca^{2+}$, $K^+$, and $Mg^{2+}$ in
most of the cities were higher than 1 (Fig. 8c and S1), indicating the effects of anthropogenic source
or crustal source. The $EF_{sea}$ for $Cl^-$ presented the lower value in many coastal cities such as Beihai
(0.53) and Haikou (0.52), while they were significantly higher than 1 in some inland cities such as
Daqing (13.11). The spatial variability of $EF_{sea}$ for $Cl^-$ confirmed the spatial difference of $Cl^-/Na^+$
between coastal cities and inland ones mentioned above. Compared with $EF_{sea}$, the $EF_{soil}$ of ions
generally displayed remarkably spatial variation. The $EF_{soil}$ of $SO_4^{2-}$, $NO_3^-$, $F^-$, and $Cl^-$ showed
notably higher values in SEC, implicating the effects of industrial activity (Fig. 8a-b and S2a-b).
The $EF_{soil}$ of $NH_4^+$ presented markedly higher value in the eastern region of Inner Mongolia and
Heilongjiang province such as Hegang (325.69) (Fig. 8c) because intensive grazing was beneficial





to the $NH_3$ emission (Kobbing et al., 2014). It was interesting to note that the $EF_{soil}$ of $Na^+$ showed
higher value in some cities around Qinghai Lake and the evaporation of salt lake could contribute
to the higher $EF_{soil}$ of $Na^+$ (Fig. S2c). The $EF_{soil}$ of crustal ions such as $Mg^{2+}$ and $K^+$ in NWC were
close to 1, reflecting the contributions of dust events and soils (Fig. S2e-f).

Based on the $EF_{sea}$ and $EF_{soil}$, the estimated SSF, CF, and AF of ions are depicted in Fig. 9, S3,

and S4. The mean SSF values of $NO_3^-$, $F^-$, $Ca^{2+}$, $NH_4^+$, $Mg^{2+}$, $K^+$, $SO_4^{2-}$, $Cl^-$, and $Na^+$ were 0%,
0.02%, 0.06%, 0.10%, 2.94%, 4.88%, 13.85%, 88.31%, and 100%, respectively. The average CF
values of $NH_4^+$, $NO_3^-$, $Cl^-$, $F^-$, $SO_4^{2-}$, $Na^+$, $K^+$, $Mg^{2+}$, and $Ca^{2+}$ reached 0.01%, 0.02%, 0.59%, 10.04%,
19.50%, 35.34%, 95.12%, 97.06%, and 99.94%, respectively. The AF value was considered to be
the contribution ratio of each ion except SSF and CF. The AF values of $Ca^{2+}$, $K^+$, $Mg^{2+}$, $Na^+$, $Cl^-$,
$SO_4^{2-}$, $F^-$, $NH_4^+$, and $NO_3^-$ reached 0%, 0%, 0%, 0%, 11.10%, 66.65%, 89.94%, 99.89%, and 99.98%,
respectively. The results suggested that $NO_3^-$, $SO_4^{2-}$, $NH_4^+$, and $F^-$ were mainly sourced from
anthropogenic activities based on minor SSF and CF. It was well documented that the combustion
of fossil fuels, iron and steel industrial emission, and vehicle exhaust were main sources of $SO_4^{2-}$
and $NO_3^-$ across China (Song et al., 2006; Yang et al., 2016). In the present study, the AF values of
$NO_3^-$ in all of cities were higher than 90%, and those of $SO_4^{2-}$ in half of the cities were higher than
60%. Besides, the utility of nitrogen fertilization, and human and livestock excretions were treated
as the main source of $NH_4^+$ emission over China (Cao et al., 2009). Herein, 82.5% of cities across
China showed the higher AF value of $NH_4^+$ (> 90%). $Ca^{2+}$, $K^+$, and $Mg^{2+}$ were mainly derived from
crustal origin based on the high CF values. Although the $K^+$ concentration in the fine particles was
usually sourced from biomass burning, the component in the coarse particles generally resulted from
the soil erosion and dust re-suspension (Cao et al., 2009). The higher CF values of $K^+$ in most of



cities in China such as Aksu (Xinjiang autonomous region) and Bayin (Gansu province) suggested
that the wet deposition has become the main removal mechanism for the $K^+$ in the coarse particles
(Lim et al., 1991). The $Na^+$ and $Cl^-$ ions were mainly originated from sea source because they were
main components of sea-salt and sea-spray aerosol (Prather et al., 2013), which was also supported
by the higher SSF value.

At a spatial scale, the highest AF values of $NO_3^-$, $SO_4^{2-}$, $NH_4^+$, and $F^-$ were mainly concentrated

on East China and SWC (Fig. 9a-c, S3a-c), which was similar to the spatial variation of population.
The emissions of aerosols and their precursors released by human activities were mainly
concentrated on East China (Fu and Chen 2016), thereby leading to high AF values of these
secondary ions. Indeed, many cities in NC such as Handan and Shijiazhuang showed the higher AF
value, which revealed the effects of power plant, non-ferrous smelting, and oral mining. The SSF
value of $Cl^-$ exhibited high value in Xinjiang and Qinghai province (i.e., Altay and Haibei), SWC
(i.e., Chengdu and Guangan) (Fig. S3d-e), and some coastal cities (i.e., Ningbo and Shanghai). The
higher SSF values of $Cl^-$ in SWC and coastal cities of East China were mainly controlled by Indian
monsoon and East Asia monsoon driven atmospheric transport, respectively (Gu et al., 2016).
However, it was assumed that the higher SSF value of $Cl^-$ in the region close to Qinghai Lake could
be linked to the evaporation of saline (Bian et al., 2017). However, the relatively higher CF value
of $Cl^-$ was centered on Ningxia autonomous region and Shaanxi province, which was frequently
exposed of Aeolian dust especially under the process of wind erosion (Lyu et al., 2017). As the
typical crustal ions, $K^+$ and $Mg^{2+}$ in the most regions of China generally showed high CF values,
especially in some cities of SWC (i.e., Guiyang, Zunyi, Zhaotong) (Fig. S4a-d). It was supposed
that the severe soil erosion and loss, and rocky desertification frequently observed in Yungui Plateau





contributed to the higher CF value in this region (Jiang et al., 2014). The SSF of $K^+$ and $Mg^{2+}$
showed high values in some coastal cities (i.e., Sanya and Ningbo), and some cities of NWC such
as Haibei (Qinghai). The evaporation of salt in East China Sea and Qinghai Lake could play a vital
role on the $K^+$ and $Mg^{2+}$ in these areas (Bian et al., 2017).
3.4.2 The FA-MLR analysis
In order to enhance the reliability of source identification, the FA method was also utilized to
identify the source of chemical compositions in the precipitation. The FA results of four seasons are
summarized in Tab. 3. Three principal components were extracted from the rainwater samples, all
of which explained 85.6% of the total variance. The Kaiser-Meyer-Olkin indicator (0.85) was higher
than 0.7, suggesting that three factors extracted in the present study was reasonable. Factor 1
grouped $NO_3^-$ , $F^-$, $NH_4^+$, and $SO_4^{2-}$, accounting for 52.3% of the variance, which was generally
associated with dense anthropogenic activities (Nayebare et al., 2016; Zhang et al., 2017b). Factor
2 displayed high loadings of $Na^+$ and $Cl^-$, indicating the effects of sea-salt and sea-spray aerosol
(Gupta et al., 2015). The result was also in good agreement with the high SSF value of $Na^+$ and $Cl^-$
supported by geochemical index method. Factor 3 occupied 9.54% of the total variance and was
dominated by $Ca^{2+}$, $Mg^{2+}$, and $K^+$. The former two ions were considered to be the important
indicators of crustal origin or windblown dust source, which were commonly stored in soils and
dusts (Kchih et al., 2015). $K^+$ was also observed in urban fugitive dusts, although it was generally
considered as an important fingerprint of biomass burning (Shen et al., 2016). As a whole, the result
of FA was in coincident with that obtained from the EF and geochemical index method.
Although the key origins were isolated via the FA method, the contribution ratio of these
sources to the water-soluble ions were still unknown. Thus, the FA-MLR method was further applied





to quantify the contribution ratio of several sources to these ions in the 320 cities over China (Fig.
10a-d). In four seasons, the mean contributions of the anthropogenic source ($NO_3^-$, $SO_4^{2-}$, $NH_4^+$, and
$F^-$: 79.10%, 46.12%, 82.40%, and 71.02%) were significantly higher than those of sea source
(13.76%, 31.71%, 11.09%, and 11.52%) and crustal origin (7.14%, 22.17%, 6.52%, and 17.46%)
for $NO_3^-$, $SO_4^{2-}$, $NH_4^+$, and $F^-$. Nevertheless, the contribution ratio was in the order of crustal origin
($K^+$, $Ca^{2+}$, and $Mg^{2+}$: 77.44%, 82.17%, and 70.51%) > anthropogenic source (13.91%, 10.20%, and
18.36%) > sea source (8.65%, 7.64%, and 11.14%) for $K^+$, $Ca^{2+}$, and $Mg^{2+}$. The sea source was the
dominant factor for the accumulation of $Na^+$ and $Cl^-$ in the rainwater, followed by the crustal origin
and the anthropogenic source. In addition, the contribution ratios of three sources showed the slight
variation in different seasons (Fig. 10). For instance, the contribution ratio of sea source to most
inorganic ions especially $Na^+$ and $Cl^-$ displayed the highest one in summer, followed by ones in
spring and autumn, and the lowest one in winter because the intense evaporation of sea salt in
summer was inclined to release more ions to the atmosphere (Teinilä et al., 2014). The contribution
ratio of anthropogenic activities presented the notable increase from summer to winter for $SO_4^{2-}$
because of dense coal combustion (20 kg coal/$m^2$) for domestic heating in winter (Zhao et al., 2016).
3.5 The deposition flux of the water-soluble ions and their key factors
At a national scale, the annually mean deposition fluxes of $NO_3^-$, $Cl^-$, $Ca^{2+}$, $K^+$, $F^-$, $NH_4^+$, $Mg^{2+}$,
$SO_4^{2-}$, and $Na^+$   over China were 13.25, 8.44, 13.80, 2.49, 1.15, 5.90, 2.27, 33.41, and 4.39 kg ha$^{-1}$
yr$^{-1}$ during 2011-2016. The deposition fluxes of $NO_3^-$, $Ca^{2+}$, $K^+$, $NH_4^+$, and $Na^+$ increased from 13.67
to 14.83 kg ha$^{-1}$ yr$^{-1}$, 13.32 to 16.99 kg ha$^{-1}$ yr$^{-1}$, 2.47 to 2.79 kg ha$^{-1}$ yr$^{-1}$, 5.21 to 6.48 kg ha$^{-1}$ yr$^{-1}$,
and 4.17 to 5.74 kg ha$^{-1}$ yr$^{-1}$ from 2011 to 2013, respectively. However, they increased to 13.65,
11.01, 2.52, 5.90, and 3.69 kg ha$^{-1}$ yr$^{-1}$ in 2016. The wet deposition fluxes of $F^-$ and $Mg^{2+}$ over China



decreased from 1.27 to 0.96 kg ha$^{-1}$ yr$^{-1}$ and 2.76 to 1.85 kg ha$^{-1}$ yr$^{-1}$ during 2012-2014, respectively.
However, they began to increase slightly to 1.17 and 2.15 in 2016, respectively. The wet deposition
fluxes of Cl$^-$ and SO$_4^{2-}$ showed gradual decrease from 9.80 and 38.87 kg ha$^{-1}$ yr$^{-1}$ to 8.09 and 26.54
kg ha$^{-1}$ yr$^{-1}$ during 2011-2016, respectively. On average, the wet deposition flux of NO$_3^-$ were higher
by 2.25 times than that of NH$_4^+$, which was in contrast to the results of the dry deposition reported
by Xu et al. (2015). All of the water-soluble ions showed the highest wet deposition fluxes in
summer, followed by ones in spring and autumn, and the lowest ones in winter, which was probably
attributed by the high washout effect due to rain in summer (Jia et al., 2014). Based on the results
of the correlation analysis, the precipitation showed the significant relationship with the deposition
fluxes of the water-soluble ions ($p < 0.05$). In addition, the wet deposition fluxes of the water-soluble
ions showed the significantly spatial variation, which were in good agreement with the spatial
distribution of the water-soluble ion concentrations except Ca$^{2+}$ (Fig. S5).

In order to determine the dominant factors affecting the wet deposition fluxes of the water-

soluble ions across China, GDP, GIP, TEC, N fertilizer use, vehicle ownership, UGS, dust days,
many meteorological factors (i.e., T$_{max}$, T$_{min}$, WS), and air pollutants (i.e., SO$_2$ and NO$_2$) were
introduced as the explanatory variables. The SR analysis results are depicted in Tab. 4. GIP, vehicle
ownership, NO$_2$, T$_{min}$, and wind speed served as the key factors affecting apparently the wet
deposition of NO$_3^-$ at a national scale. The atmospheric emission of NO$_x$ from coal-fired power
plants was estimated about 7489.6 kt in 2010, although many newly built power plants were
equipped with advanced low NO$_x$ burner (LNB) systems (Tian et al., 2013). Zhang et al. (2014)
estimated that NO$_x$ from vehicle emissions reached 4570 kt in 2008, which was considered as the
second NO$_x$ source only to industrial activities. The NO$_x$ released from anthropogenic activity could



enhance the $NO_2$ concentration in the ambient air, which could be also transformed to $NO_3^-$ via
oxidation in the atmosphere, especially under the condition of high temperature and low WS (Zhang
et al., 2016). The wet deposition of $NH_4^+$ were affected by N fertilizer use, UGS, and $NO_2$ over
China. Russel et al. (1998) recommended early that $NH_4^+$ in the precipitation was most likely
derived from the N fertilizer use via an isotope techniques coupled with back trajectory analysis.
Besides, Teng et al. (2017) demonstrated that the emission from UGS was identified to contribute
to the atmospheric $NH_3$ significantly during 60% of the sampling times, which could increase the
$NH_4^+$ concentration in the precipitation due to the photochemical reaction. The wet deposition flux
of $SO_4^{2-}$ was closely associated with TEC in the 320 cities of China, respectively. It was supposed
that the $SO_2$ emission were dependent on the use of coal and petroleum (Lu et al., 2010). While
terrestrial petroleum emissions have declined in recent years, the emissions from international
shipping have offset the decrease of terrestrial petroleum (Smith et al., 2011). In the present study,
the deposition of some crustal ions were linked to the dust days because they were mainly derived
from the dust storm or soil (Deshmukh et al., 2011; Zhang et al., 2011). The $F^-$ deposition was
associated with GIP due to the contributions of the coal-fired power plant fly ash and industrial raw
material (Kong et al., 2011).

The GWR method was used to calculate the local regression coefficients in order to determine

the dominant factor affecting the deposition of the water-soluble ions at the regional scale (Fig. 11
and S6). The mean $R^2$ of GWR method was 0.50 over China, and the p value was lower than 0.05,
which suggested that the GWR method could be applicable to the study.The local regression
coefficient of dust days for crustal ions including $Ca^{2+}$, $Cl^-$, $K^+$, and $Mg^{2+}$ increased from SEC to
NWC (Fig. S6a-e), suggesting that dust days played a significant role on the crustal ions in NWC





due to high intensity of dust deposition and extremely high WS (Zhang et al., 2017a). The influence
of GIP on the $F^-$ and $NO_3^-$ increased from West China to East China, and displayed the higher value
in some cities of YRD (i.e., Shanghai, Hangzhou) because many coal-fired power plants, cement
plants, and municipal solid waste incineration plants were located in YRD (Hua et al., 2016; Tian et
al., 2012; Tian et al., 2014) (Fig. S6f and 11a). The influence of N fertilizer use on $NH_4^+$ was
concentrated on some cities of NEC such as Jiamusi (Heilongjiang province) (Fig. 11b-c), Harbin
(Heilongjiang province), Changchun (Jilin province) because the largest commodity grain base were
located in Heilongjiang and Jilin province, leading to the higher N fertilizer use (Cheng and Zhang,
2005). In contrast to the effects of GIP, the TEC influence increased gradually from SEC to NWC,
and showed the highest value in Xinjiang autonomous region (i.e., Altay) (Fig. 11d). It has been
demonstrated that an inverted U-shaped curve (Environment Kuznets Curve) between per capita
GDP and energy consumption was generally observed during the development of economy (Song
et al., 2013; Yang et al., 2017). The Environment Kuznets Curve denoted that the energy
consumption displayed positive relationship with per capita GDP in the early stage of development.
However, the positive relationship tended to transform into the negative relevance with the
development of economy because the reliance on the energy-intensive industries would be reduced
in the developed stage (Yang et al., 2017). It was assumed that Xinjiang autonomous region kept at
the early stage of the inverted-U curve and largely rested on the energy-intensive industries as the
less-developed province (Yang et al., 2017). However, some developed provinces in SEC such as
Zhejiang and Jiangsu have sped up structural transformation of the economy and reduce the reliance
on the heavy industries. The influence of UGS and vehicle ownership peaked in Shandong province
(i.e., Qingdao, Jinan) and YRD (i.e., Shanghai, Hangzhou) (Fig. 11e-f). It was supposed that the



UGS and vehicle ownership in these cities showed higher values among all of the 320 cities
(National Bureau of Statistics of China). Apart from the effects of socioeconomic factors, the
meteorological factors also played significant roles on $NO_3^-$. The influences of air temperature and
WS both increased from East China to West China, and showed the highest values in Xinjiang
province (Fig. 11g-h). Zhang et al. (2017a) demonstrated that the strong dust events along with high
WS contributed to the neutralization of $NO_3^-$, although the $NO_2$ concentrations in some cities of
Xinjiang province were significantly higher than other regions of China.
**4.    Conclusions**

This study newly reported spatiotemporal variation of nine water-soluble ions in the

precipitation across the whole China during 2011-2016. The mean pH and EC values varied
significantly compared with those during 1980-2000 because the implementation of special air
pollution control measures have mitigated the air pollution in China. The concentrations of $Na^+$,
$NO_3^-$, and $SO_4^{2-}$ increased from 7.26 ± 2.51, 11.56 ± 3.71, and 33.73 ± 7.59 μeq/L to 11.04 ± 4.64,
13.59 ± 2.63, and 41.95 ± 8.64 μeq/L during 2011 and 2014, while they decreased from the highest
ones in 2014 to 9.75 ± 2.89, 12.29 ± 4.02, and 30.57 ± 7.43 μeq/L in 2016, respectively. The
concentrations of $Ca^{2+}$, $NH_4^+$, and $Mg^{2+}$ increased by 86.26%, 178.50%, and 19.71% from 2011 to
2013, whereas they decreased from 58.84 ± 10.31, 41.33 ± 10.26, and 10.49 ± 3.07 in 2013 to 31.20
± 8.48, 18.13 ± 4.84, and 8.93 ± 2.92 μeq/L in 2016, respectively. The concentration of $F^-$ decreased
linearly by 5.58%/yr during 2012-2016. The mean concentrations of $SO_4^{2-}$, $NO_3^-$ and $F^-$ showed the
highest values in winter, followed by ones in spring and autumn, and the lowest ones in summer. It
was supposed that the dense anthropogenic activities such as domestic combustion for heating and
adverse meteorological conditions. The crustal ions ($Ca^{2+}$, $Mg^{2+}$, and $K^+$) peaked in spring and



summer, suggesting the contributions of fugitive dusts. The $Na^+$ and $Cl^-$ were markedly affected by
evaporation of sea salt. All of the water-soluble ions in the precipitation exhibited notably spatial
variability. The secondary ions ($SO_4^{2-}$, $NO_3^-$ and $NH_4^+$), and $F^-$ peaked in YRD (i.e., Changzhou,
Hangzhou, and Nanjing) owing to the intensive energy consumption and industrial activities. The
higher S content in the coal and unfavorable diffusion conditions contributed to the higher
concentrations of secondary ions in SB (i.e., Chengdu, Leshan, and Dazhou). The crustal ions and
sea-salt ions showed the highest concentrations in semi-arid regions (i.e., Guyuan, Jiayuguan) and
coastal cities (i.e., Qingdao, Lianyungang), respectively.
The EF method, geochemical index method, and FA-MLR method consistently suggested that
$NO_3^-$ , $F^-$, $NH_4^+$, and $SO_4^{2-}$ were dominated by anthropogenic activities. However, the $Na^+$ and $Cl^-$
were closely associated with sea-salt aerosol. $Ca^{2+}$, $Mg^{2+}$, and $K^+$ were mostly derived from crustal
source. The SR analysis and GWR method implied that GIP, TEC, vehicle ownership, and N
fertilizer use played the important roles on $SO_4^{2-}$, $NO_3^-$, $NH_4^+$, and $F^-$. However, the crustal ions
were significantly affected by dust events. The correlation between influential factors and the ions
in the wet deposition showed significantly spatial variability. The influence of dust days on the
crustal ions increased from SEC to NWC, whereas the influence of socioeconomic factors on
secondary ions showed the highest value in East China.
The present study validate the model estimations of the water-soluble ions deposition at a
national scale, and provide the fundamental data for the prevention and control of acid deposition
and air pollution. However, there were several plausible contributors to the uncertainty. First of all,
the monitoring sites were distributed unevenly and relatively scarce sites were located in Northwest
China. Moreover, the limited independent variables were included into the models. Thus, further





studies were required to establish more representative monitoring sites and incorporate more
variables to reduce the uncertainty associated with the ions deposition.
**Acknowledgements**
This work was supported by National Key R&D Program of China (2016YFC0202700), National
Natural Science Foundation of China (Nos. 91744205, 21777025, 21577022, 21177026),
International cooperation project of Shanghai municipal government (15520711200), and Marie
Skłodowska-Curie Actions (690958-MARSU-RISE-2015). The meteorological data are avaiable at
http://data.cma.cn/. The socioeonomic data are collected from http://www.stats.gov.cn/.





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



# Figure and table caption

**Fig. 1** The spatial distribution of 320 cities and five ecological regions.

**Fig. 2** The inter-annual and seasonal variation of pH and EC of the precipitation in China.

**Fig. 3** The spatial distribution of pH and EC of the precipitation in China.

**Fig. 4** The temporal variation of water-soluble ions in the precipitation.

**Fig. 5** The spatial variation of $NO_3^-$, $NH_4^+$, and $SO_4^{2-}$ in the precipitation.

**Fig. 6** The spatial distribution of $Ca^{2+}$, $Cl^-$, $F^-$, $K^+$, $Mg^{2+}$, and $Na^+$ in the precipitation.

**Fig. 7** The triangular diagrams of NF for main alkaline ions.

**Fig. 8** The $EF_{sea}$ and $EF_{soil}$ of $NO_3^-$, $SO_4^{2-}$, and $NH_4^+$.

**Fig. 9** The spatial variation of SSF, CF, and AF for $NO_3^-$, $NH_4^+$, and $SO_4^{2-}$ in the precipitation.

**Fig. 10** The seasonal difference of contribution ratios of anthropogenic source, crustal source, and, sea source.

**Fig. 11** The local regression coefficient of influential factors for the $NO_3^-$, $NH_4^+$, and $SO_4^{2-}$.

**Tab. 1** The comparison of physicochemical properties and chemical composition in the precipitation.

**Tab. 2** The mean enrichment factor relative to sea and soil, and the source contribution (%) of major ions in China (SSF denotes sea salt fraction, CF represents the crustal source, AF indicates the anthropogenic fraction).

**Tab. 3** The loading matrix of precipitation in four seasons of China.

**Tab. 4** The results of stepwise regression method.





**Fig. 1**

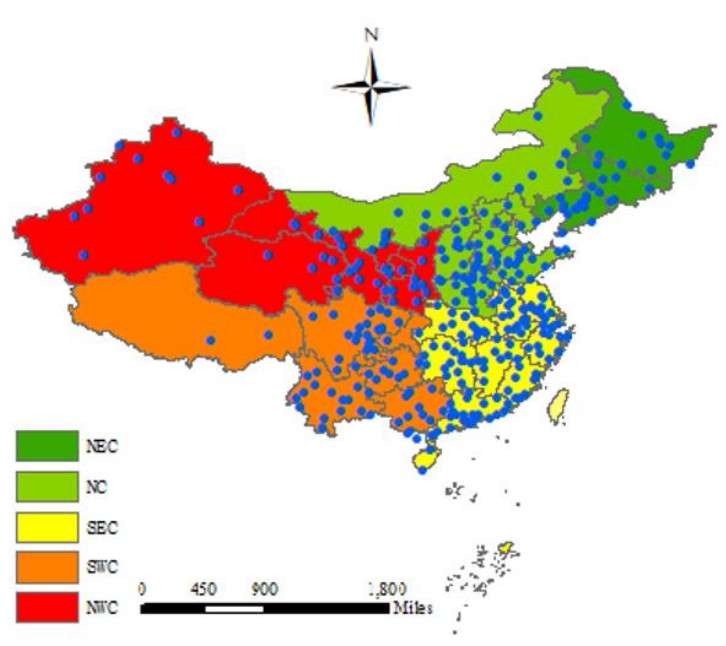





**Fig. 2**

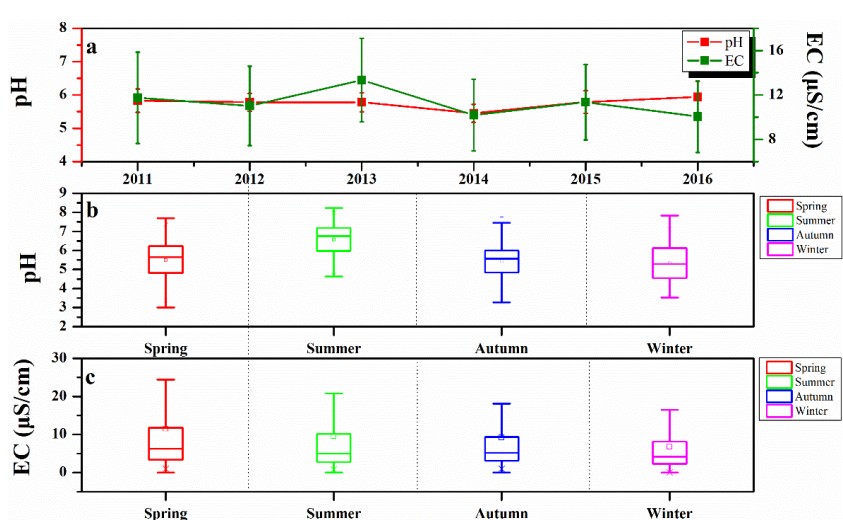





**Fig. 3**

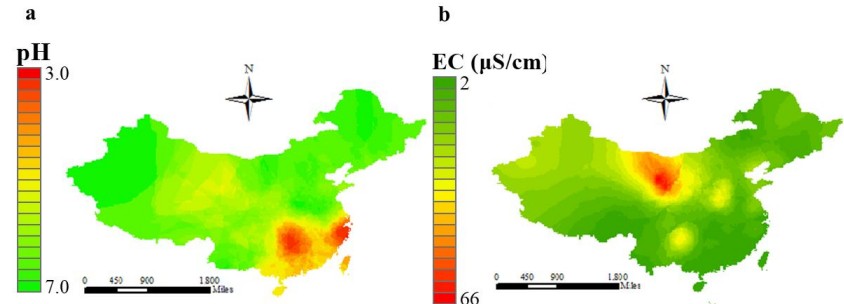





**Fig. 4**

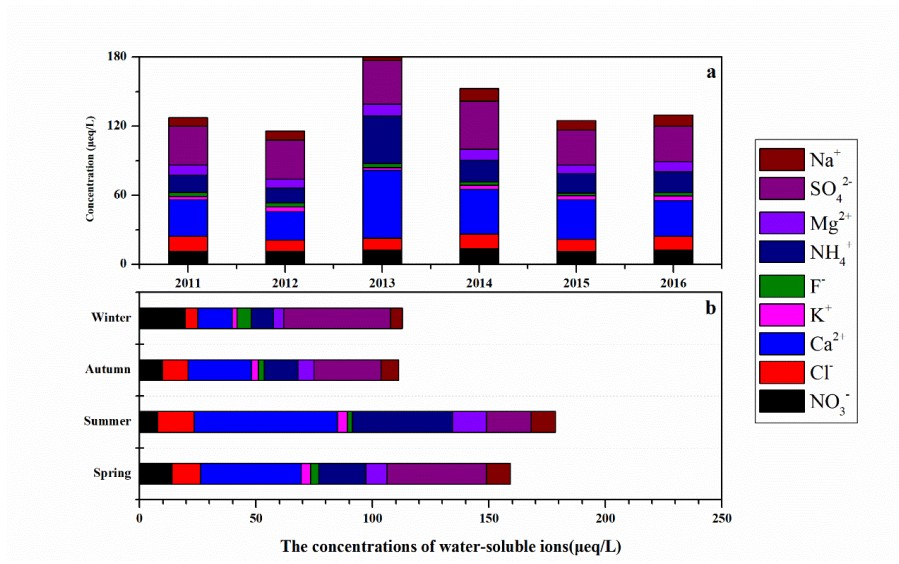



**Fig. 5**

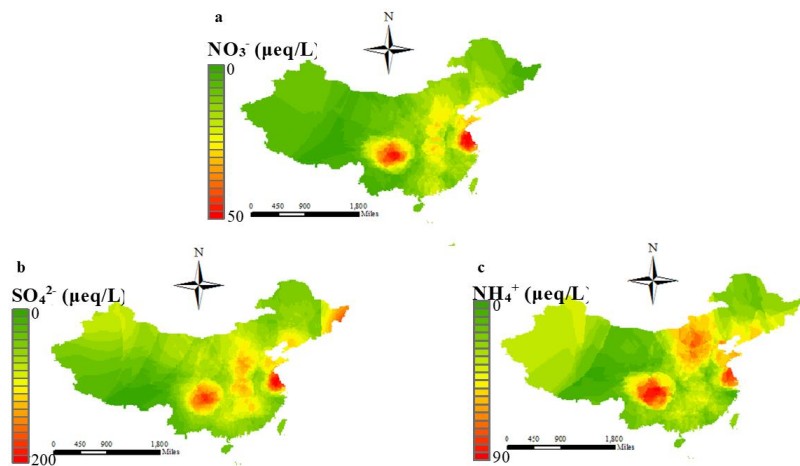





**Fig. 6**

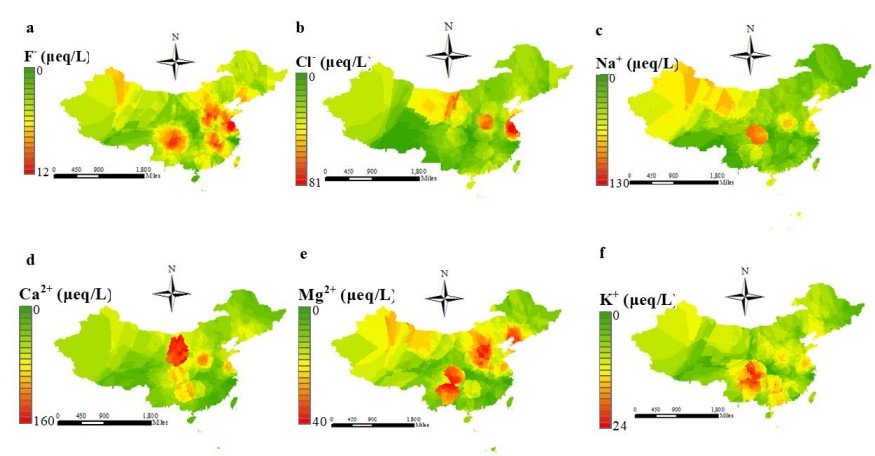





**Fig. 7**

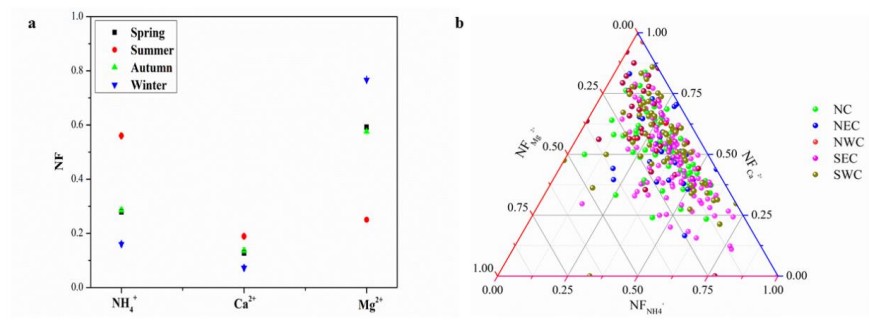





**Fig. 8**

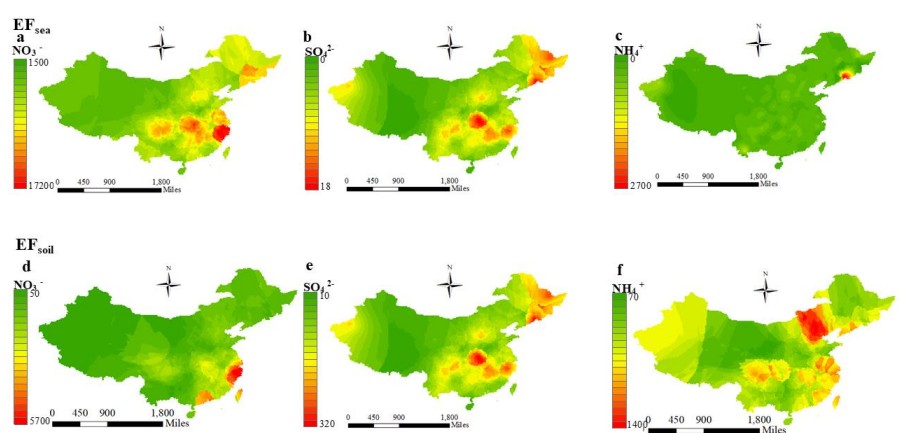





**Fig. 9**

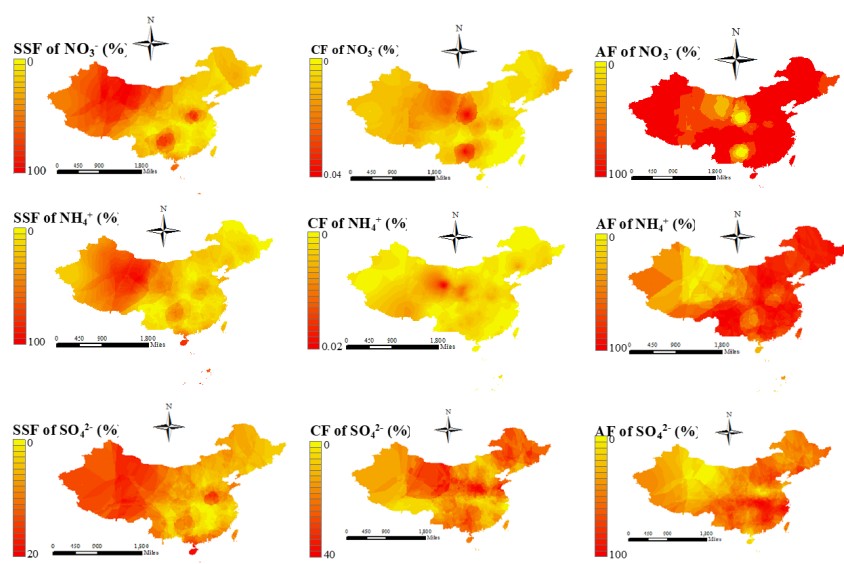





**Fig. 10**

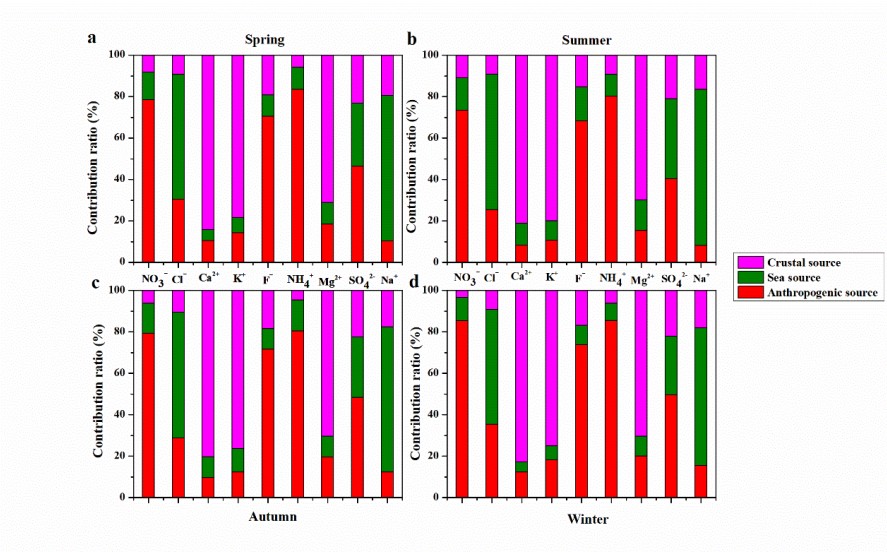



**Fig. 11**

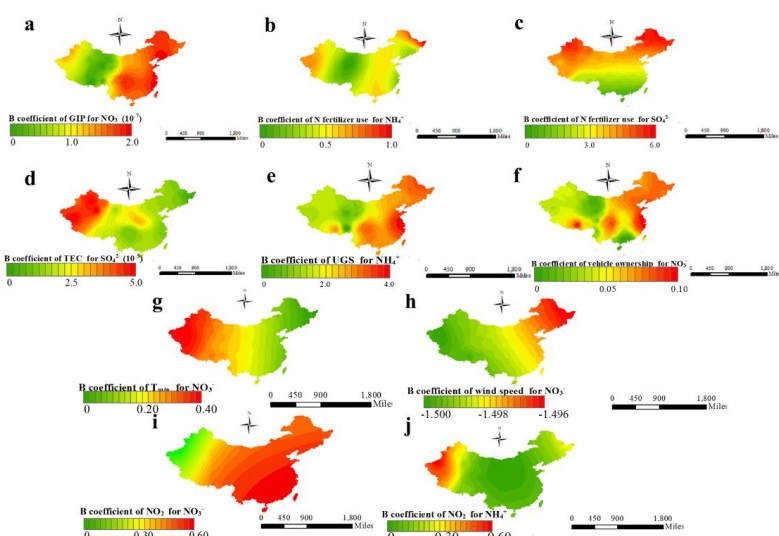



**Tab. 1**

| | pH | EC | $NO_3^-$ | $Cl^-$ | $Ca^{2+}$ | $K^+$ | $F^-$ | $NH_4^+$ | $Mg^{2+}$ | $SO_4^{2-}$ | $Na^+$ | Year | References |
|---|---|---|---|---|---|---|---|---|---|---|---|---|---|
| Beijing | 5.68 | 9.89 | 15.13 | 6.62 | 26.27 | 1.80 | 2.24 | 45.33 | 5.51 | 31.28 | 3.39 | 2011- | This study |
| Zhengzhou | 6.09 | 26.44 | 37.10 | 72.45 | 109.23 | 8.25 | 5.80 | 23.82 | 20.54 | 25.80 | 6.40 | 2011- | This study |
| Harbin | 6.13 | 7.41 | 9.87 | 20.71 | 21.98 | 5.02 | 5.03 | 11.96 | 9.55 | 28.76 | 22.00 | 2011- | This study |
| Shenyang | 5.76 | 8.40 | 24.52 | 15.90 | 75.32 | 2.59 | 4.32 | 40.68 | 22.68 | 57.57 | 16.88 | 2011- | This study |
| Qingdao | 5.32 | 16.53 | 5.25 | 5.79 | 28.18 | 2.07 | 1.34 | 9.28 | 9.80 | 10.96 | 25.30 | 2011- | This study |
| Shanghai | 4.39 | 2.50 | 40.06 | 4.15 | 19.09 | 1.07 | 1.45 | 17.48 | 4.71 | 29.13 | 20.36 | 2011- | This study |
| Wuhan | 4.68 | 2.66 | 11.61 | 2.12 | 13.55 | 0.76 | 1.07 | 9.38 | 2.63 | 27.93 | 1.28 | 2011- | This study |
| Guangzhou | 4.98 | 2.84 | 26.74 | 19.38 | 41.60 | 9.42 | 3.93 | 13.58 | 8.33 | 35.76 | 9.57 | 2011- | This study |
| Chengdu | 4.89 | 6.03 | 48.08 | 22.13 | 44.42 | 12.60 | 9.21 | 65.19 | 8.23 | 77.16 | 15.06 | 2011- | This study |
| Lhasa | 5.21 | 4.51 | 0.50 | 1.65 | 7.66 | 0.48 | 0.94 | 0.91 | 1.28 | 1.44 | 1.62 | 2011- | This study |
| Urumqi | 6.13 | 13.41 | 16.87 | 30.38 | 115.24 | 4.76 | 2.02 | 73.76 | 19.41 | 56.76 | 28.87 | 2011- | This study |
| Lanzhou | 5.05 | 58.06 | 16.19 | 4.93 | 51.84 | 1.24 | 1.57 | 3.05 | 8.17 | 33.30 | 10.87 | 2011- | This study |
| Jiuzhaigou | 5.95 | 15.70 | 9.10 | 44.10 | 55.80 | 34.80 | 0.86 | 18.40 | 5.60 | 15.90 | 12.60 | 2015- | Qiao et al. (2018) |
| Yulong | 5.94 | 10.30 | 4.00 | 1.96 | 37.7 | 2.46 | 1.20 | 13.20 | 5.68 | 28.30 | 3.72 | 2012 | Niu et al. (2014) |
| Nam Co | 6.59 | 19.70 | 10.00 | 19.20 | 301 | 14.50 | - | 18.10 | 7.43 | 15.50 | 15.40 | 2005 | Li et al. (2007) |
| Southern | - | - | 20.97 | 31.06 | 46.68 | 11.14 | - | 58.57 | 22.55 | 45.97 | 56.41 | 2005- | Tsai et al. (2011) |
| Petra, | 6.80 | 160 | 35.70 | 80.60 | 163.10 | 26.30 | - | 18.40 | 62.30 | 53.20 | 75.60 | 2002- | Al-Khashman et al. (2005) |
| Tokyo, | 4.52 | - | 30.50 | 55.20 | 24.90 | 2.90 | - | 40.4 | 11.5 | 50.2 | 37.0 | 1990- | Okuda et al. (2005) |
| Guaíba, | 5.92 | 10.8 | 4.00 | 13.80 | 21.50 | 5.81 | 5.90 | 38.90 | 8.85 | 23.10 | 15.10 | 2002 | Migliavacca et al. (2005) |
| Sao Paulo, | - | - | 15.60 | 0.90 | 5.50 | 3.70 | - | 27.90 | 1.70 | 8.60 | 3.60 | 2000 | Fornaro and Gutz (2003). |
| Singapore | - | - | 16.80 | 22.10 | 21.7 | 3.96 | - | 17.3 | 7.46 | 58.7 | 31.1 | 1997- | Balasubramanian et al. (2001) |
| Newark, | - | - | 14.40 | 10.70 | 6.00 | 1.30 | - | 24.40 | 3.30 | 38.10 | 10.90 | 2006- | Song and Gao (2009) |
| Patras, | 5.16 | -- | 19.40 | 114.30 | 98.50 | 6.60 | -- | 16.30 | 30.40 | 46.10 | 90.20 | 2000- | Glavas and Moschonas (2002) |
| Sardinia, | 5.18 | -- | 29 | 322 | 70 | 17 | -- | 25 | 77 | 90 | 252 | 1992- | Le Bolloch and Guerzoni (1995) |
| Adirondack, | 4.50 | -- | 22.60 | 2.14 | 3.59 | 0.33 | -- | 10.50 | 0.99 | 36.90 | 1.61 | 1988- | Ito et al. (2002) |



**Tab. 2**

|  | EF$_{sea}$ | EF$_{soil}$ | SSF | CF | AF |
|---|---|---|---|---|---|
| NO$_3^-$ | 3507.49 | 59.36 | 0 | 0.02 | 99.98 |
| Cl- | 1.13 | 169.88 | 88.31 | 0.59 | 11.10 |
| Ca$^{2+}$ | 231.56 | 1.00 | 0.06 | 99.94 | 0 |
| K$^+$ | 16.16 | 0.83 | 4.88 | 95.12 | 0 |
| F$^-$ | 5864.28 | 9.96 | 0.02 | 10.04 | 89.94 |
| NH$_4^+$ | 10.51 | 86.31 | 0.10 | 0.01 | 99.89 |
| Mg$^{2+}$ | 10.18 | 0.55 | 2.94 | 97.06 | 0 |
| SO$_4^{2-}$ | 7.22 | 5.13 | 13.85 | 19.50 | 66.65 |
| Na$^+$ | 1.00 | 1.83 | 64.66 | 35.34 | 0 |



**Tab. 3**

| Season | Variable | F1 | F2 | F3 |
|---|---|---|---|---|
| Overall | $NO_3^-$ | **0.71** | 0.24 | 0.45 |
| | $Cl^-$ | 0.43 | **0.64** | -0.12 |
| | $Ca^{2+}$ | 0.42 | -0.22 | **0.75** |
| | $K^+$ | 0.39 | 0.18 | **0.72** |
| | $F^-$ | **0.68** | -0.20 | 0.45 |
| | $NH_4^+$ | **0.74** | 0.35 | 0.13 |
| | $Mg^{2+}$ | -0.41 | 0.10 | **0.66** |
| | $SO_4^{2-}$ | **0.63** | 0.23 | 0.14 |
| | $Na^+$ | -0.02 | **0.65** | 0.45 |
| Spring | $NO_3^-$ | **0.76** | 0.11 | -0.32 |
| | $Cl^-$ | -0.33 | **0.59** | 0.26 |
| | $Ca^{2+}$ | 0.32 | -0.16 | **0.80** |
| | $K^+$ | -0.36 | 0.06 | **0.78** |
| | $F^-$ | **0.70** | -0.10 | 0.20 |
| | $NH_4^+$ | **0.68** | 0.29 | -0.46 |
| | $Mg^{2+}$ | -0.38 | 0.42 | **0.69** |
| | $SO_4^{2-}$ | **0.77** | 0.31 | 0.22 |
| | $Na^+$ | -0.04 | **0.72** | 0.46 |
| Summer | $NO_3^-$ | **0.63** | 0.24 | -0.33 |
| | $Cl^-$ | 0.42 | **0.66** | -0.38 |
| | $Ca^{2+}$ | 0.44 | -0.26 | **0.85** |
| | $K^+$ | -0.37 | 0.19 | **0.70** |
| | $F^-$ | **0.54** | -0.32 | 0.48 |
| | $NH_4^+$ | **0.59** | 0.33 | -0.47 |
| | $Mg^{2+}$ | 0.32 | -0.38 | **0.60** |
| | $SO_4^{2-}$ | **0.56** | 0.36 | 0.34 |
| | $Na^+$ | -0.09 | **0.75** | 0.49 |
| Autumn | $NO_3^-$ | **0.73** | -0.14 | 0.38 |
| | $Cl^-$ | -0.39 | **0.62** | 0.29 |
| | $Ca^{2+}$ | 0.32 | -0.16 | **0.80** |
| | $K^+$ | 0.45 | -0.09 | **0.68** |
| | $F^-$ | **0.68** | -0.15 | 0.28 |





| | | | | |
|---|---|---|---|---|
| | $NH_4^+$ | **0.69** | 0.42 | -0.45 |
| | $Mg^{2+}$ | -0.29 | 0.32 | **0.71** |
| | $SO_4^{2-}$ | **0.68** | -0.29 | 0.23 |
| | $Na^+$ | -0.14 | **0.69** | -0.37 |
| Winter | $NO_3^-$ | **0.79** | 0.23 | -0.36 |
| | $Cl^-$ | -0.38 | **0.49** | 0.29 |
| | $Ca^{2+}$ | 0.39 | -0.35 | **0.65** |
| | $K^+$ | -0.39 | 0.08 | **0.72** |
| | $F^-$ | **0.75** | 0.08 | -0.24 |
| | $NH_4^+$ | **0.73** | 0.26 | -0.42 |
| | $Mg^{2+}$ | 0.35 | -0.49 | **0.75** |
| | $SO_4^{2-}$ | **0.79** | 0.22 | 0.36 |
| | $Na^+$ | -0.16 | **0.54** | 0.33 |

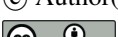



**Tab. 4**

| Dependent variables | Independent variables | Partial regression coefficients | $R^2$ | t value | p value |
|---|---|---|---|---|---|
| $NO_3^-$ | GIP | $8.42\times10^{-8}$ | 0.62 | 4.03 | 0.00 |
| | Vehicle ownership | 0.03 | | -2.39 | 0.01 |
| | $NO_2$ | 0.34 | | 4.29 | 0.00 |
| | $T_{min}$ | 0.15 | | 1.34 | 0.02 |
| | Wind speed | -1.49 | | -1.69 | 0.03 |
| $Cl^-$ | Dust days | 0.12 | 0.52 | 2.14 | 0.04 |
| $Ca^{2+}$ | $PM_{10}$ | 0.36 | 0.56 | 3.26 | 0.00 |
| | Dust days | 132.74 | | 2.99 | 0.00 |
| $K^+$ | Dust days | 2.09 | 0.49 | 2.03 | 0.02 |
| $F^-$ | GIP | $0.54\times10^{-7}$ | 0.50 | 2.31 | 0.02 |
| $NH_4^+$ | N fertilizer use | 0.14 | 0.48 | 2.46 | 0.02 |
| | UGS | $1.33\times10^{-4}$ | | 1.79 | 0.04 |
| | $NO_2$ | 0.25 | | 1.98 | 0.03 |
| $Mg^{2+}$ | Dust days | 2.36 | 0.43 | 1.65 | 0.05 |
| $SO_4^{2-}$ | TEC | $2.80\times10^{-5}$ | 0.64 | 3.07 | 0.00 |
| | N fertilizer use | 3.36 | | 3.59 | 0.00 |
| $Na^+$ | Dust days | 2.46 | 0.46 | 1.69 | 0.04 |