# Peer review of "The wet deposition of the inorganic ions in the 320 cities across 2 China: spatiotemporal variation, source apportionment, and dominant factors"

_Atmospheric Chemistry and Physics, 2019_

## Referee Comment (RC1) · Anonymous Referee #1 · 22 May 2019

Deposition of inorganic ions is an important indicator of air pollutant emissions and has potentially large impact on ecosystem. Attributed to its large size and complicated sources of atmospheric components, China is of big diversity on inorganic ion deposition and it is great challenge to quantify the spatial and temporal patterns of deposition. Based on intensive sampling and chemical analysis at sites across the country, this work presents informative results on wet deposition of ions, and analyzed the seasonal and annual changes in deposition. The sources of the deposition were evaluated as well based on specific statistic or arithmetic methods. In general, the paper is of comprehensive information and well organized. Before it can be accepted for publication, however, some issues should be further stressed or discussed, and certain information

needs to be clarified as well. Details follow.

Major comment: 1. Section 2.1: sampling site. One of the most valuable contributions of this work is the sampling and chemical analysis at a great number of cities and sites across the country. However, the strategy of the site selection is unclear. How many sites are located in urban and how many are in remote/suburban regions? Such kind of information is helpful for audience to judge the representativeness of the sampling.

2. Section 2.2: Regarding the sampling, it is unclear whether the sampling covers the whole studying period for all of the sites? Or the sampling period varied by site? If so, what's the reason? Moreover, the frequency of sampling collection should also be described.

3. Section 2.5, what's the purpose of this section? Was the method applied for the spatial pattern of wet deposition? Is it related with the spatial interpolation? The method should be explained more carefully.

4. Lines 285-294, Section 3.1, the author stated the decreasing trend of SO2 and NOx emissions resulting in the increased pH for the studying period. In Section 3.2, they presented that the peak sulfate and nitrate peaked in 2014, which seems contradicting to the inter-annual variation of SO2 and NOx emissions. Could you explain the possible reasons?

5. Lines 383-385. This statement might not necessarily true for China, as coal burning and some industrial sources are also very important sources of NOx. Vehicle cannot dominate the growth of NOx emissions and thereby NO3- concentrations in precipitation. Moreover, what do you mean by "linearly increase"?

6. Section 3.2.2. The seasonal variation of sulfate and nitrate concentrations in precipitation could also be influenced by some other factors. For example, if high temperature in summer elevated oxidation of precursors, how could it result in smaller concentrations? Is it possible that more abundant rainfall dilute the concentrations? Moreover,

heating in south China is not as frequent as in north. Here I suggest the authors make a more detailed classification of sampling sites and check the difference between north and south China, and that between rural and urban sites.

7. Line 482. The Ca was extremely high in summer, but the dust emissions might not be high in summer due to precipitation. I guess there are some other reasons besides those mentioned by the authors.

8. Lines 664 and 665. Ca emissions could also from some coal burning and industry sources. That means anthropogenic sources could contribute to Ca. I feel that the uncertainty of the method should be discussed here, as you indicate that the contribution from human activities was almost zero for Ca. Moreover, over one third of sulfate was expected to come from natural sources (AF=66.65%), what are they?

9. Minor issues: Lines 216-217, do the "rain" and "precipitation" mean the same thing in eqs (8) and (9)? Line 223, what is FA? Line 288, Liu or Lu? Line 297-298, why compared with 2000? Should it be 2010? Line 741 increased or decreased? Lines 842-843, rewrite the sentence. It is not clear.

---

## Author Comment (AC1) · 31 May 2019

Dear Professor Joshua Fu: Here we submit our revised manuscript for consideration to be published on Atmospheric Chemistry and Physics The further information about our manuscript is as follows: Topic: The wet deposition of the inorganic ions in the 320 cities across China: spatiotemporal variation, source apportionment, and dominant factors Type of Manuscript: article Authors: Rui Lia, Lulu Cuia, Yilong Zhaoa, Ziyu Zhanga, Tianming Suna, Junlin Lia, Wenhui Zhoua, Ya Menga, Kan Huanga, Hongbo Fua,b,c * *Corresponding author: Hongbo Fu; Address: Department of Environmental Science and Engineering, Fudan University, Shanghai 200433,

[Figure]

China; Tel.: (+86)21-5566-5189; Fax: (+86)21-6564-2080; Email: fuhb@fudan.edu.cn

Firstly, we acknowledge the suggestions of editor and anonymous reviewers, and are also grateful to your efficient serving. We have updated the manuscript on the basis of these valuable comments. Our responses were listed as following: Reviewer #1: Deposition of inorganic ions is an important indicator of air pollutant emissions and has potentially large impact on ecosystem. Attributed to its large size and complicated sources of atmospheric components, China is of big diversity on inorganic ion deposition and it is great challenge to quantify the spatial and temporal patterns of deposition. Based on intensive sampling and chemical analysis at sites across the country, this work presents informative results on wet deposition of ions, and analyzed the seasonal and annual changes in deposition. The sources of the deposition were evaluated as well based on specific statistic or arithmetic methods. In general, the paper is of comprehensive information and well organized. Before it can be accepted for publication, however, some issues should be further stressed or discussed, and certain information needs to be clarified as well. Details follow. Comment 1: Section 2.1: sampling site. One of the most valuable contributions of this work is the sampling and chemical analysis at a great number of cities and sites across the country. However, the strategy of the site selection is unclear. How many sites are located in urban and how many are in remote/suburban regions? Such kind of information is helpful for audience to judge the representativeness of the sampling. Response: Thank for reviewer's suggestion. (Line 151-152) Indeed, the information about the sampling sites is helpful for reader. Therefore, we have added the detailed description about the sampling sites. The strategy of the site selection is to assure that the monitoring sites in each city were a mixture of urban sites and suburban/rural sites, which can accurately reflect the acid deposition status of each city. In the present study, 850 monitoring sites were located in urban areas and 432 sites were distributed on the rural regions. Comment 2: Section 2.2: Regarding the sampling, it is unclear whether the sampling covers the whole studying period for all of the sites? Or the sampling period varied by site? If so, what's the reason?

Moreover, the frequency of sampling collection should also be described. Response: Thank for reviewer's suggestion. (Line 159-162) All of the samples were collected in all of the monitoring sites simultaneously. Sampling collection frequency was strongly dependent on the rain event, and each sample was properly collected during the precipitation event when the wet-only deposition instrument was under the normal condition. Comment 3: Section 2.5, what's the purpose of this section? Was the method applied for the spatial pattern of wet deposition? Is it related with the spatial interpolation? The method should be explained more carefully. Response: Thank for reviewer's suggestion. (Section 2.5) The GWR model was not related with the spatial interpolation, but reflected the spatial correlation of socioeconomic factors and inorganic ion depositions. The model was to explore the effects of socioeconomic factors on wet deposition of inorganic ions in consideration of the spatial correlation. Compared with the traditional multiple regression analysis, GWR incorporated the spatial weight matrix into the novel model because the inorganic ion deposition fluxes for neighboring cities generally showed the significantly spatial correlation. Furthermore, GWR model can investigate the spatial variability of the correlation between socioeconomic factors and inorganic ion deposition fluxes compared with the multiple regression analysis. Comment 4: Lines 285-294, Section 3.1, the author stated the decreasing trend of SO2 and NOx emissions resulting in the increased pH for the studying period. In Section 3.2, they presented that the peak sulfate and nitrate peaked in 2014, which seems contradicting to the inter-annual variation of SO2 and NOx emissions. Could you explain the possible reasons? Response: Thank for reviewer's suggestion. (Line 272-305) Indeed, we stressed that the SO2 and NOx emissions in most regions of China displayed the decrease during 2011-2016 compared with those before 2000, which led to the higher pH value compared with those before 2000. The result was drawn based on the data comparison with previous studies. However, it did not mean the pH value over China exhibited the linear increase during 2011-2016. Actually, the pH value increased from 2011 to 2014, while it decreased from the peak to the lower value in 2016. Meanwhile, both of the sulfate and nitrate

displayed the similarly annual variations with the pH value. It might be attributed to that these acidic ions might be not very sensitive to the emission decrease. Therefore, it was not contradictory. Comment 5: Lines 383-385. This statement might not necessarily true for China, as coal burning and some industrial sources are also very important sources of NOx. Vehicle cannot dominate the growth of NOx emissions and thereby NO3- concentrations in precipitation. Moreover, what do you mean by "linearly increase"? Response: I agree with reviewer's suggestion. (Line 390-393) Indeed, the vehicle cannot dominate the growth of NOx emission. Based on the reference review, we found that the annual trend of nitrate in the precipitation was in good agreement with the ambient NO2 level. It suggested that stricter controls on NOx emissions from power plants might be counteracted by the increase of power plants and energy consumption (Liu et al. 2015a; Wang et al. 2018). "Linearly increase" meant that the vehicle emissions displayed the gradual increase since 1998 and the trend was similar to the straight line. It reflected that the NOx emission from vehicle exhaust exhibited the rapid increase during the past decades. Although the increase of vehicle volume played an important role on the nitrate in the precipitation, the increase of power plants and energy consumption might be more important. Comment 6: Section 3.2.2. The seasonal variation of sulfate and nitrate concentrations in precipitation could also be influenced by some other factors. For example, if high temperature in summer elevated oxidation of precursors, how could it result in smaller concentrations? Is it possible that more abundant rainfall dilute the concentrations? Moreover, heating in south China is not as frequent as in north. Here I suggest the authors make a more detailed classification of sampling sites and check the difference between north and south China, and that between rural and urban sites. Response: Thank for reviewer's suggestion. (Line 459-486) Indeed, the higher temperature in summer promoted the oxidation of precursors to sulfate and nitrate, while the dense rainfall could scavenge and washout, the particles and then decrease the concentrations of sulfate and nitrate. We agreed with reviewer suggestion. We have classified all of the cities into South and North China and rural and urban sites. Overall, the acidic ions in both of North

China and South China exhibited the higher concentrations in winter and spring, and the lower ones in summer. However, the NO3- concentration in South China displayed a slight difference, which showed the highest one in spring. It was assumed that the relatively scarce precipitation in spring could be responsible for the higher NO3- level. Comment 7: Line 482. The Ca was extremely high in summer, but the dust emissions might not be high in summer due to precipitation. I guess there are some other reasons besides those mentioned by the authors. Response: Thank for reviewer's suggestion. (Line 490) Based on the reference review, many previous studies have found that the Ca2+ in the precipitation was higher in summer compared with other seasons (Niu et al., 2014). It was widely acknowledged that soil-derived crust particulates in the atmosphere were deposited concurrent with the initial rainfall events occurring in summer. Indeed, the dust emission from desert was fewer in summer, while the road dust cannot be neglected. Lyu et al. (2016) demonstrated that the high temperature coupled with strong wind caused the lower water content in the road, leading to higher tendency of road dust re-suspension in the Wuhan summer. Comment 8: Lines 664 and 665. Ca emissions could also from some coal burning and industry sources. That means anthropogenic sources could contribute to Ca. I feel that the uncertainty of the method should be discussed here, as you indicate that the contribution from human activities was almost zero for Ca. Moreover, over one third of sulfate was expected to come from natural sources (AF=66.65%), what are they? Response: I agree with reviewer's suggestion. (Line 722-727) Indeed, there are some uncertainties in the geochemical index method, and thus we have discussed the uncertainty in the last paragraph of section 3.4.1. First of all, the background values of Na+ in the sea and Ca2+ in the soil displayed the higher uncertainty, which varied significantly with the study areas. Unfortunately, the background values of Na+ and Ca2+ over China were absent. Besides, the source classification might be not very accurate because many other sources such as forest fire and volcanic eruption were also ignored. The sulfate generally possesses some natural sources including the contributions of sea-spray, dust emission, forest fire, and volcanic eruption. Comment 9: Minor issues: Lines

216-217, do the "rain" and "precipitation" mean the same thing in eqs (8) and (9)? Line 223, what is FA? Line 288, Liu or Lu? Line 297-298, why compared with 2000? Should it be 2010? Line 741 increased or decreased? Lines 842-843, rewrite the sentence. It is not clear. Response: Thank for reviewer's suggestion. The "rain" and "precipitation" mean the same thing. To avoid the misunderstanding, we have replaced the rain by precipitation. FA means factor analysis. Line 294-295, the Liu has been revised to Lu. Line 304-305, we compared the pH value with that in 2000 rather than 2010. It was assumed that few references concerned about the pH value over China. To date, we only found a paper about the pH value before 2000, and thus we compare with the pH value with that in 2000, and explore the factors for the pH difference. Line 766, the "increased" has been replaced by "decreased". Line 867-868: The sentences has been replaced by "The results of SR analysis and GWR method implied that GIP, TEC, vehicle ownership, and N fertilizer use were main factors for $SO_4^{2-}$, $NO_3^-$, $NH_4^+$, and $F^-$ in the precipitation".

Please also note the supplement to this comment:
https://www.atmos-chem-phys-discuss.net/acp-2019-52/acp-2019-52-AC1-supplement.zip
* * *

---

## Referee Comment (RC2) · Anonymous Referee #2 · 13 Jun 2019

I noticed that the authors improved their manuscript substantially after careful revision based on reviewer's comments. I am satisfied at their revision. The paper now can be accepted for publication in ACP as its current form.

---

## Author Comment (AC2) · 14 Jun 2019

Dear Professor Joshua Fu: Here we submit our revised manuscript for consideration to be published on Atmospheric Chemistry and Physics The further information about our manuscript is as follows: Topic: The wet deposition of the inorganic ions in the 320 cities across China: spatiotemporal variation, source apportionment, and dominant factors Type of Manuscript: article Authors: Rui Lia, Lulu Cuia, Yilong Zhaoa, Ziyu Zhanga, Tianming Suna, Junlin Lia, Wenhui Zhoua, Ya Menga, Kan Huanga, Hongbo Fua,b,c * *Corresponding author: Hongbo Fu; Address: Department of Environmental Science and Engineering, Fudan University, Shanghai 200433,

China; Tel.: (+86)21-5566-5189; Fax: (+86)21-6564-2080; Email: fuhb@fudan.edu.cn
Firstly, we acknowledge the suggestions of editor and anonymous reviewers, and are also grateful to your efficient serving. We have updated the manuscript on the basis of these valuable comments. Our responses were listed as following: Reviewer #1: Comment: I noticed that the authors improved their manuscript substantially after careful revision based on reviewer's comments. I am satisfied at their revision. The paper now can be accepted for publication in ACP as its current form. Response: Thank for reviewer's suggestion. I have uploaded the clean manuscript to the system.

Please also note the supplement to this comment:
https://www.atmos-chem-phys-discuss.net/acp-2019-52/acp-2019-52-AC2-supplement.zip

---

## Author Response (AR1)

**Dear Professor Joshua Fu:**

Here we submit our revised manuscript for consideration to be published on **Atmospheric Chemistry and Physics**

The further information about our manuscript is as follows:

**Topic:** The wet deposition of the inorganic ions in the 320 cities across China: spatiotemporal variation, source apportionment, and dominant factors

**Type of Manuscript:** article

**Authors:** Rui Li[a], Lulu Cui[a], Yilong Zhao[a], Ziyu Zhang[a], Tianming Sun[a], Junlin Li[a], Wenhui Zhou[a], Ya Meng[a], Kan Huang[a], Hongbo Fu[a,b,c] *

 ***Corresponding author:**

Hongbo Fu; Address: Department of Environmental Science and Engineering, Fudan University, Shanghai 200433, China; Tel.: (+86)21-5566-5189; Fax: (+86)21-6564-2080; Email: fuhb@fudan.edu.cn

Firstly, we acknowledge the suggestions of editor and anonymous reviewers, and are also grateful to your efficient serving. We have updated the manuscript on the basis of these valuable comments. Our responses were listed as following:

**Reviewer #1:**

**Comment:** I noticed that the authors improved their manuscript substantially after careful revision based on reviewer's comments. I am satisfied at their revision. The paper now can be accepted for publication in ACP as its current form.

**Response:** Thank for reviewer's suggestion. I have uploaded the clean manuscript to the system.

[revised manuscript text omitted]